# Structural and mechanistic basis of the EMC-dependent biogenesis of distinct transmembrane clients

Lakshmi E Miller-Vedam[1,2,3,4†], Bastian Bräuning[5†], Katerina D Popova[3,4,6†], Nicole T Schirle Oakdale[4‡], Jessica L Bonnar[3,4], Jesuraj R Prabu[5], Elizabeth A Boydston[4§], Natalia Sevillano[7#], Matthew J Shurtleff[4¶], Robert M Stroud[2], Charles S Craik[7], Brenda A Schulman[5*], Adam Frost[2*], Jonathan S Weissman[3,4,8*]

[1]Molecular, Cellular, and Computational Biophysics Graduate Program, University of California, San Francisco, San Francisco, United States; [2]Department of Biochemistry and Biophysics, University of California, San Francisco, San Francisco, United States; [3]Department of Biology, Whitehead Institute, MIT, Cambridge, United States; [4]Department of Cellular and Molecular Pharmacology, University of California, San Francisco, San Francisco, United States; [5]Department of Molecular Machines and Signaling, Max Planck Institute of Biochemistry, Martinsried, Germany; [6]Biomedical Sciences Graduate Program, University of California, San Francisco, San Francisco, United States; [7]Department of Pharmaceutical Chemistry, University of California, San Francisco, San Francisco, United States; [8]Howard Hughes Medical Institute, Chevy Chase, United States

*For correspondence:
schulman@biochem.mpg.de
(BAS);
adam.frost@ucsf.edu (AF);
weissman@wi.mit.edu (JSW)

†These authors contributed equally to this work

Present address: ‡Gilead Sciences, Foster City, United States; §Whitehead Institute, MIT, Cambridge, United States; #FairJourney Biologics, Porto, Portugal; ¶Lycia Therapeutics, South San Francisco, San Francisco, United States

Competing interests: The authors declare that no competing interests exist.

**Abstract** Membrane protein biogenesis in the endoplasmic reticulum (ER) is complex and failure-prone. The ER membrane protein complex (EMC), comprising eight conserved subunits, has emerged as a central player in this process. Yet, we have limited understanding of how EMC enables insertion and integrity of diverse clients, from tail-anchored to polytopic transmembrane proteins. Here, yeast and human EMC cryo-EM structures reveal conserved intricate assemblies and human-specific features associated with pathologies. Structure-based functional studies distinguish between two separable EMC activities, as an insertase regulating tail-anchored protein levels and a broader role in polytopic membrane protein biogenesis. These depend on mechanistically coupled yet spatially distinct regions including two lipid-accessible membrane cavities which confer client-specific regulation, and a non-insertase EMC function mediated by the EMC lumenal domain. Our studies illuminate the structural and mechanistic basis of EMC's multifunctionality and point to its role in differentially regulating the biogenesis of distinct client protein classes.

## Introduction

Integral membrane proteins serve diverse and critical cellular roles, including signal transduction, lipid biosynthesis, adhesion, and transport of molecules across the bilayer. In eukaryotic cells, the endoplasmic reticulum (ER) serves as the primary site of integral membrane protein synthesis, targeting (co- or post-translationally), insertion, folding, and quality control (*Ellgaard et al., 2016*; *Costa et al., 2018*). However, the features of membrane-spanning regions (e.g. low hydrophobicity, charged residues, non-optimal lengths, lipid- and ion-binding sites and hairpins or kinked transmembrane helices) that mediate important functions pose particular challenges for transmembrane

**eLife digest** Cells are surrounded and contained by a plasma membrane consisting of a double layer of fats and proteins. These proteins monitor and facilitate the movement of food, oxygen and messages in and out of the cell, and help neighboring cells communicate. Membrane proteins are manufactured in a cell compartment called the endoplasmic reticulum. Cellular machines called ribosomes visit this compartment's membrane to manufacture proteins that need to be secreted or embedded into the cell's membranes. As these proteins are made, they are pulled into the endoplasmic reticulum so they can be folded correctly and inserted in the membrane. A cellular machine in this compartment's membrane that aids this process is the endoplasmic reticulum membrane protein complex (EMC). Many steps can go wrong during protein assembly, so to control protein quality, the EMC has to accommodate the variety of complex physical features that proteins can have.

To explore the activity of the EMC, Miller-Vedam, Bräuning, Popova et al. studied the normal structure of the EMC in both yeast and human cells grown in the lab. These snapshots of the complex in different species had a lot in common, including how the complex was arranged within and around the membrane.

Next, Miller-Vedam, Bräuning, Popova et al. generated 50 mutant versions of the EMC in human cells to determine how changing different parts of the complex affected the production of three proteins that rely on the EMC to fold correctly. These proteins were an enzyme called squalene synthase, a signaling protein called the beta adrenergic receptor and sigma intracellular receptor 2, a protein involved in the regulation of cholesterol levels.

Mutations in the section of the EMC outside of the endoplasmic reticulum, within the main cellular compartment, negatively impacted the stability of squalene synthase. This section of the EMC provides a platform where proteins can associate before entering the membrane.

The part of EMC that spans the membrane contains both a fat-filled cavity and a cavity with a 'door' that is either open or closed. Mutations in this section disrupted the insertion of both squalene synthase and the beta adrenergic receptor into the membrane, a role performed by the cavity with the door. The specific role of the fat-filled cavity is still not fully understood, but a mutation affecting this cavity disrupts the correct production of all three proteins studied.

The largest section of the complex, which sits inside the endoplasmic reticulum, protected proteins as they folded, ensuring they were not destroyed for being folded incorrectly before they were fully formed. Mutations in this part of the EMC negatively impacted the stability of sigma intracellular receptor 2 without negatively affecting the other proteins.

This molecular dissection of the activity of the EMC provides insights into how membrane proteins are manufactured, stabilized, coordinated, and monitored for quality. These findings could contribute towards the development of new treatments for certain congenital diseases. For example, cystic fibrosis, retinitis pigmentosa, and Charcot-Marie-Tooth disease are all thought to be caused by mutations within membrane proteins that require the EMC during their production.

protein biosynthesis and folding. Consequently, membrane protein biogenesis is prone to failure, and this can lead to cellular stress and disease (*Marinko et al., 2019*). Thus, it is important to understand the cellular factors that facilitate proper membrane protein biogenesis for such challenging clients.

The ER membrane protein complex (EMC) has emerged as a conserved player in the process of membrane protein biogenesis. It was first identified in *Saccharomyces cerevisiae* as an abundant and stable multi-protein membrane complex whose disruption results in stress mirroring that caused by misfolded membrane proteins (*Jonikas et al., 2009*). Loss of the EMC in mammalian cells is associated with failed biogenesis and degradation of a subset of membrane proteins (*Christianson et al., 2012*). Accordingly, the EMC has been implicated in several mechanistically distinct steps of membrane protein biogenesis, stabilization, and quality control (*Bircham et al., 2011*; *Richard et al., 2013*; *Satoh et al., 2015*; *Savidis et al., 2016*; *Shurtleff et al., 2018*; *Volkmar et al., 2019*; *Tian et al., 2019*).

One well-established EMC function is as an insertase for terminal transmembrane helices. EMC's insertase function has been demonstrated for two classes of clients: low hydrophobicity tail-anchored proteins (i.e. those that contain C-terminal membrane anchors) and a subset of polytopic transmembrane proteins in which the first helix is inserted with the N-terminus in the lumen (*Guna et al., 2018*; *Chitwood et al., 2018*). However, many studies indicate EMC functions beyond initial insertion of N- or C-terminal helices. The EMC has been implicated in the biogenesis and stability of many membrane protein classes that do not require a terminal transmembrane insertase (*Bircham et al., 2011*; *Louie et al., 2012*; *Richard et al., 2013*; *Shurtleff et al., 2018*; *Coelho et al., 2019*; *Luo et al., 2002*; *Volkmar et al., 2019*; *Talbot et al., 2019*; *Petkovic et al., 2020*). Recent studies have shown that the EMC is required for stability of internal transmembrane helices of human and viral multi-pass membrane proteins (*Hiramatsu et al., 2019*; *Lin et al., 2019*; *Ngo et al., 2019*; *Coelho et al., 2019*; *Xiong et al., 2020*). Additionally, the human EMC (hEMC) physically interacts with the NS4A-B region of the Dengue Virus polyprotein following Sec61-dependent translocation and signal peptidase cleavage, suggesting roles in post-translational stabilization of polytopic membrane proteins (*Ngo et al., 2019*; *Lin et al., 2019*). Similarly, the *S. cerevisiae* EMC (yEMC) co-immunoprecipitated with full-length polytopic transmembrane clients, including Pma1p (*Luo et al., 2002*), Mrh1p, and Fks1p (*Shurtleff et al., 2018*). In addition to varying types of transmembrane protein clients, the EMC also associates with a range of regulatory factors, including many general and substrate-specific chaperones in the cytoplasm and in the ER lumen (*Bagchi et al., 2016*; *Coelho et al., 2019*; *Kudze et al., 2018*; *Richard et al., 2013*; *Shurtleff et al., 2018*).

The complex architecture of the EMC provides additional support for multifunctionality in membrane protein biogenesis. The EMC is an eight (yeast) or nine (mammalian) component, 248–284 kDa complex with considerable mass in the ER lumen, membrane, and cytosol. The cytoplasmic domain contains conserved tetratricopeptide repeats (TPR) repeats in EMC2, and the human complex accommodates an additional subunit, EMC8/9, whose function is not yet understood. The ER lumenal domain in yeast does not contain an N-terminal EMC1 expansion seen in hEMC. Notably, the ER lumenal domain has been linked to a number of disease-associated phenotypes (*Junes-Gill et al., 2011*; *Probert et al., 2015*; *Harel et al., 2016*; *Abu-Safieh et al., 2013*; *Diamantopoulou et al., 2017*; *Marquez et al., 2020*), and presents the possibility of additional functions for the human lumenal domain. One EMC subunit (EMC3) shares limited sequence homology with a family of insertases that are evolutionarily related to the bacterial insertase YidC (*Samuelson et al., 2000*; *Kumazaki et al., 2014*; *Borowska et al., 2015*; *Anghel et al., 2017*), perhaps explaining the insertase function of the complex. During the preparation of our manuscript, studies describing the structures of the yeast Get1/Get2/Get3 structures, human WRB/CAML/TRC40 (*McDowell et al., 2020*), translocon bound to Nicalin-TMEM147-NOMO (*McGilvray et al., 2020*), human structures of the EMC (*O'Donnell et al., 2020*; *Pleiner et al., 2020*), and the yeast structure of the EMC (*Bai et al., 2020*) were published. Those studies focused on the insertase activities of these proteins from the individual species; however, the elaboration of the EMC compared to other known membrane protein biogenesis factors and a diverse client range points to additional functionality that has so far eluded mechanistic explanation. Notably, a systematic structure-based functional analysis across species, conformations, the three distinct EMC domains, and including non-insertase client proteins and mutagenesis of the extensive lumenal domain had not been done.

Here, we determined high-resolution cryo-EM structures of yeast EMC bound to a Fab and two conformations of the human EMC structure. Furthermore, we characterized the phenotypes of three distinct classes of EMC clients associated with a series of structure-based EMC mutants. Both yEMC and hEMC structures reveal a path for transmembrane helix insertion from the cytoplasm into the membrane via a conserved cavity. Our structures and mutants also revealed a second lipid-filled cavity with regions of importance for all three client types probed. Analysis of human disease mutations in hEMC1 and our structure-informed mutations enabled us to decouple the EMC insertase function from non-insertase functions and reveal a potential role of the EMC in differentially controlling the biogenesis of distinct classes of client proteins. These structure-function studies collectively establish that the EMC adopts a modular architecture enabling its diverse functions in membrane protein biogenesis.

## Results

### Overview of strategy to comprehensively reveal EMC structure and function

To comprehensively dissect both conserved and species-specific functions of the EMC, we developed approaches to produce EMC for structure determination and broad mutational analysis (*Figure 1A–D*). We developed systems to produce robust quantities of pure intact yEMC and hEMC to determine structures for the two organisms in which different facets of EMC function have been described in detail (*Jonikas et al., 2009*; *Christianson et al., 2012*; *Guna et al., 2018*; *Shurtleff et al., 2018*). Parallel efforts converged on an approach involving FLAG affinity-tagging of the EMC5 C-terminus, which was performed for endogenous yEMC and recombinant hEMC in human embryonic kidney (HEK) cells (*Figure 1—figure supplements 1–2*, *Supplementary file 1*).

In parallel, to enable testing of hypotheses based on structures, we created a suite of human (K562) knockout cell lines deleted for individual hEMC subunits - hEMC1 (lumen), hEMC2 (cytoplasm), hEMC3, and hEMC5 (transmembrane) - and a series of reporters of EMC-dependent transmembrane protein biogenesis (*Figure 1—figure supplements 3–4*). Reintroduction of the wild-type hEMC subunits in the respective knockout cells fully rescued the knockout phenotype (*Figure 1—figure supplements 5–6*). This allowed for introduction of structure-based mutations in hEMC subunits into the respective knockout cells to determine features supporting biogenesis of fluorescently tagged versions of three different types of EMC clients: the transmembrane domain of a C-terminal tail-anchored transmembrane protein (squalene synthase, SQS$^{378-410}$) (*Guna et al., 2018*), a polytopic transmembrane protein that depends on the EMC N-terminal insertase activity (Beta 1 adrenergic receptor, B1AR) (*Chitwood et al., 2018*), and a polytopic transmembrane protein (Sigma intracellular receptor 2, TMEM97) whose biogenesis requires the Sec61 translocon but does not require a terminal helix insertase (*Figure 1—figure supplements 3–6*). Three individual EMC clients were fused to mCherry fluorescent protein and GFP separated by a P2A ribosomal skipping sequence. Translation of the described mRNA generates two products due to peptide bond skipping at the P2A sequence. For each molecule of the client-mCherry fusion, there is one GFP molecule. Reduction in mCherry levels relative to GFP reflects post-translational degradation of the client fused to mCherry. Each of the client reporters were introduced into five separate cell lines: wild-type K562 cells, hEMC1 knockout K562 cells, hEMC2 knockout K562 cells, hEMC3 knockout K562 cells, and hEMC5 knockout K562 cells. Monitoring the effect of an hEMC mutation on fluorescent reporter levels provided a quantitative measure of its impact on EMC-dependent biogenesis of each class of client protein. A number of mutations of varying severity, varying conservation between yeast and human (*Figure 1—figure supplements 7–8*), were designed and tested spanning the hEMC structure. Subsequently, these 49 mutations were mapped onto the structure grouped by reporter phenotype (*Video 1*, *Supplement File 2*). To allow for direct comparison of our structure-guided mutant phenotypes with those published recently by others (*Pleiner et al., 2020*; *Bai et al., 2020*; *O'Donnell et al., 2020*), we summarized all mutant data (*Supplementary file 3*). A subset of the mutant cell lines was validated by genotyping (*Figure 1—figure supplement 9*). Western blots against the endogenous hEMC subunits allowed us to control for mutational effects on the production and stability of the hEMC complex itself. We concurrently blotted for three clients, SQS, TMEM97, and BCAP31, to assay changes in endogenous protein levels for each of the mutations (*Figure 1—figure supplements 5–6*, *Supplementary file 4*). This strategy thus distinguishes effects resulting from a global disruption of the EMC complex from those caused by specific disruption of EMC function. These functional assays of the hEMC show a broad dependence of all these clients on the EMC, consistent with previous work (*Shurtleff et al., 2018*; *Guna et al., 2018*; *Chitwood et al., 2018*; *Volkmar et al., 2019*; *Tian et al., 2019*). In order to understand the mechanism of action, we will now go in more detail through several of the mutants with the strongest functional phenotypes in differing regions of the three-dimensional structure.

### The EMC is an intricate molecular machine spanning the ER membrane and exhibits a conserved core architecture

We determined structures of yEMC and hEMC — all showing overall compositional similarity, with regional conformational differences between the yeast and human complexes (*Figure 2A–D*). We

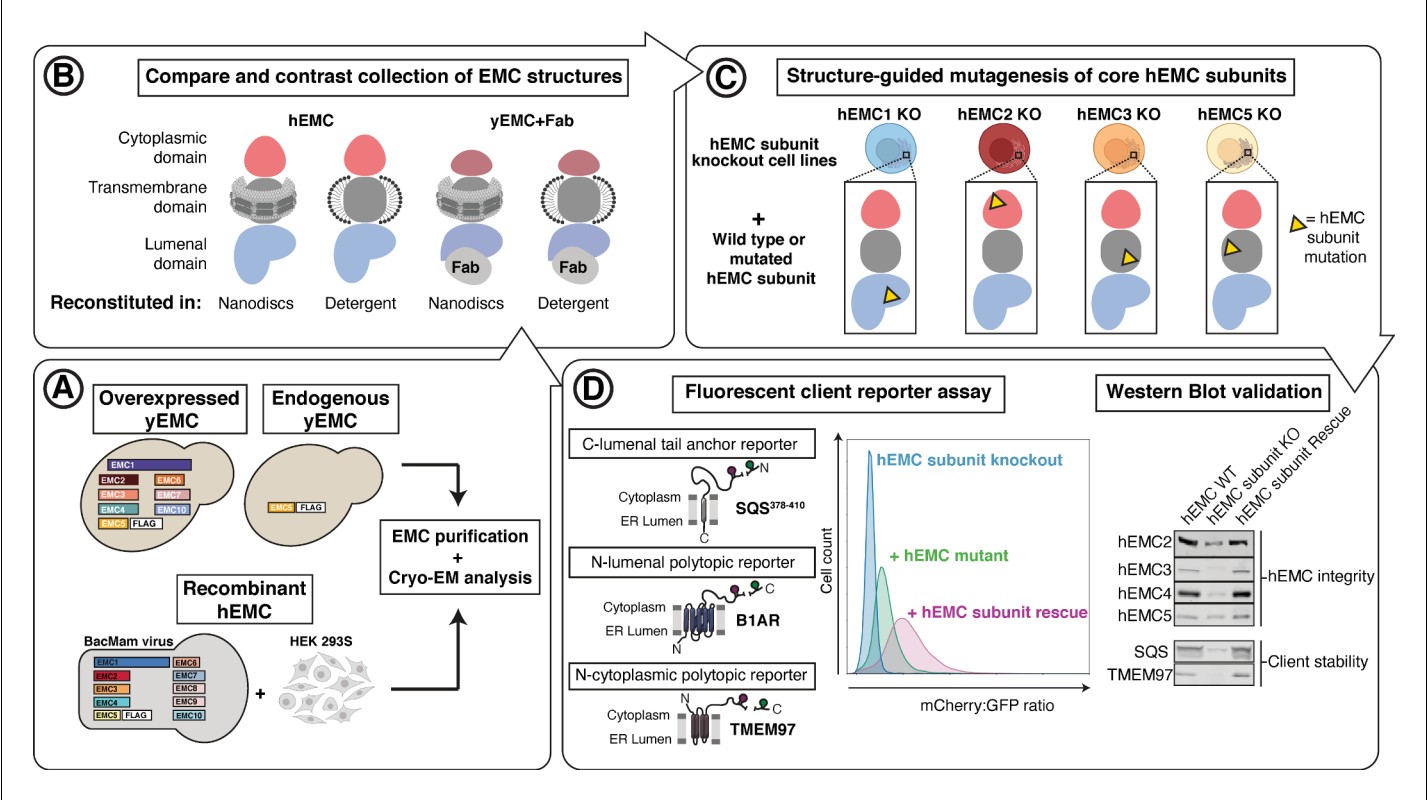

**Figure 1.** Experimental strategy for the dissection of EMC function. Schematic representation of the combined structural and mutational approach to dissect EMC function. (**A**) yEMC was purified either by overexpression of all subunits together and affinity pulldown with 3xFlag-tagged yEMC5 or by pulldown of endogenous yEMC proteins using an affinity pulldown with 3xFlag-tagged yEMC5. For hEMC, all subunits were overexpressed together with Flag-tagged EMC5 via a single recombinant BacMam virus. Both yEMC and hEMC were purified by column chromatography and subjected to cryo-EM analysis. (**B**) The obtained collection of cryo-EM structures of yEMC and hEMC in lipid nanodiscs or detergent micelles were compared to identify similarities and differences. (**C**) Structure-guided mutagenesis was performed across four core hEMC subunits: hEMC1, hEMC2, hEMC3, and hEMC5 in mammalian K562 cells. (**D**) Each hEMC subunit knockout (KO) cell line was individually transduced with three different fluorescent client reporters: SQS$^{378-410}$, full-length B1AR, and full-length TMEM97. Mutant hEMC subunits were then introduced into the corresponding subunit KO cell lines carrying each of the three fluorescent hEMC client reporters. hEMC client stability in each mutant hEMC subunit cell line was assessed by quantifying the mCherry-to-GFP ratio. Western blotting was performed for each mutant-transduced cell line to assess EMC integrity (by immunoblotting for hEMC subunits) as well as client stability (by immunoblotting for hEMC clients) compared against both wild-type (WT) and KO cell lines.

The online version of this article includes the following figure supplement(s) for figure 1:

**Figure supplement 1.** Purification of yEMC.
**Figure supplement 2.** Purification of recombinant hEMC.
**Figure supplement 3.** Fluorescent reporter cell line generation.
**Figure supplement 4.** Overview of functional assays.
**Figure supplement 5.** Western blots for EMC1 and EMC2.
**Figure supplement 6.** Western blots for EMC3 and EMC5.
**Figure supplement 7.** Amino acid conservation of EMC1.
**Figure supplement 8.** Amino acid conservation of EMC2, EMC3, EMC5.
**Figure supplement 9.** Genotyping of 10 mutants.

obtained reconstructions of yEMC bound to an antigen binding fragment (Fab) and hEMC reconstituted both in detergent micelles and lipid nanodiscs, with the latter strategy yielding the most isotropic and highest resolution data. For yEMC+FabDH4 and hEMC, the global map resolutions reached 3.2 Å and 3.4 Å, respectively (***Table 1***, ***Figure 2—figure supplements 1–4***). The cryo-EM maps allowed for de novo model building of both human and yeast complexes (***Figure 2—figure supplements 5*** and ***6***). As described in the following sections, our multiple EMC structures enable a broad survey of its conserved architecture, with variations between the structures pointing to

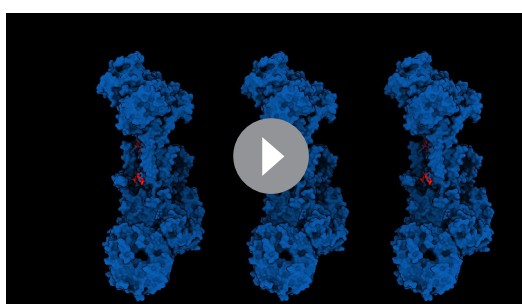

**Video 1.** hEMC mutagenesis displayed on hEMC structures. Three identical copies of hEMC in nanodisc (colored blue) are displayed here. Subsequent labeling and coloring of mutated residues by flow cytometric measure of reporter abundance, grouped into three categories: increased reporter levels (mCherry > GFP signal, colored white), wild-type levels (mCherry signal is close to GFP signal, colored light blue), and decreased reporter levels (mCherry < GFP signal, colored gold). Left hEMC structure displays phenotypes for the C-lumenal tail anchor reporter (GFP-P2A-mCherry-SQS$^{378-410}$-opsin). Middle hEMC structure displays phenotypes for the N-lumenal polytopic reporter (B1AR-mCherry-P2A-GFP). Right hEMC structure displays phenotypes for the N-cytoplasmic polytopic reporter (TMEM97-mCherry-P2A-GFP). Mutations with little to no phenotype are displayed as similar to wild-type levels.

https://elifesciences.org/articles/62611#video1

conformational and compositional differences (*Figure 2—figure supplements 7–8*). We note that our maps and models are consistent with recent cryo-EM data from yeast EMC (*Bai et al., 2020*), human EMC (*O'Donnell et al., 2020*; *Pleiner et al., 2020*), and a crystal structure of human EMC2-EMC9 (*O'Donnell et al., 2020*; *Figure 2—figure supplement 9*).

The EMC is comprised of cytoplasmic, transmembrane, and lumenal domains arranged similarly for yeast and human, despite significant evolutionary separation (*Figure 2A–B*). For both species, subunits encompassing EMC2 to EMC7 form an interconnected core complex, while there is additional density capping both the cytoplasmic and lumenal domains of hEMC, occupied by an hEMC8/9 and an hEMC1 N-terminal expansion, respectively (*Figure 2C–D*). hEMC8 and hEMC9 are paralogs of each other, which have not been identified in yeast (*Wideman, 2015*). We modeled and depict only hEMC8 for clarity, but due to the 44% sequence identity with hEMC9 and both being present in the recombinant system we refer to this as hEMC8/9. The large hEMC1 insertion in hEMC constitutes the majority of a membrane distal beta-propeller domain protruding into the lumen, a feature missing from *S. cerevisiae*. Compared to other ER-resident proteins implicated in membrane protein biogenesis (*Suloway et al., 2009*; *Pfeffer et al., 2017*; *Ramírez et al., 2019*; *McDowell et al., 2020*; *McGilvray et al., 2020*), the arrangement of domains of the EMC is unusual with the transmembrane domain connecting prominent cytoplasmic and lumenal domains (*Figure 2E*). On a global level, the structure suggests complexities beyond those of some other ER machineries fulfilling select functions in transmembrane protein biogenesis.

## The cytoplasmic domain provides a platform for protein-protein interactions

The exterior interface of the cytoplasmic domain is formed by EMC2, EMC3, EMC4, and parts of hEMC8/9 (in human), while parts of EMC5, EMC2, and EMC8/9 are shielded from the cytoplasm (*Figure 3A–B*). The helical fold of EMC2 constitutes the central organizer of this platform, established by five or six TPR motifs in human versus yeast, respectively (*Figure 3C*). TPR domains are commonly found mediating protein-protein interactions and are present in numerous well-characterized chaperone-protein and other interaction networks (*Blatch and Lässle, 1999*; *Scheufler et al., 2000*; *Schlegel et al., 2007*; *Assimon et al., 2015*; *Krysztofinska et al., 2017*; *Graham et al., 2019*). Yeast EMC2 features a more curved helical arrangement with N- and C-terminal domains in closer proximity to each other than seen in hEMC2. Notably, the canonical peptide-binding TPR groove is occupied by the partially helical C-terminus of EMC5, which forms a large interaction surface with EMC2. To test the functional roles of this interaction, we mutated three residues within the hEMC2 TPR motif (hEMC2$^{K125E + R126D + K127E}$) or a single hEMC5 residue buried in the TPR-binding groove (hEMC5$^{F90A}$). The mutations on both sides of the interface decreased hEMC integrity by western blot, with a modest decrease of hEMC subunits for hEMC5$^{F90A}$ and a strong reduction in the levels of several hEMC subunits for hEMC2$^{K125E + R126D + K127E}$ (*Figure 3C*, *Figure 3—figure supplements 1–2*, *Figure 1—figure supplements 5–6*). This suggests that this interface might be critical for EMC complex assembly rather than EMC function.

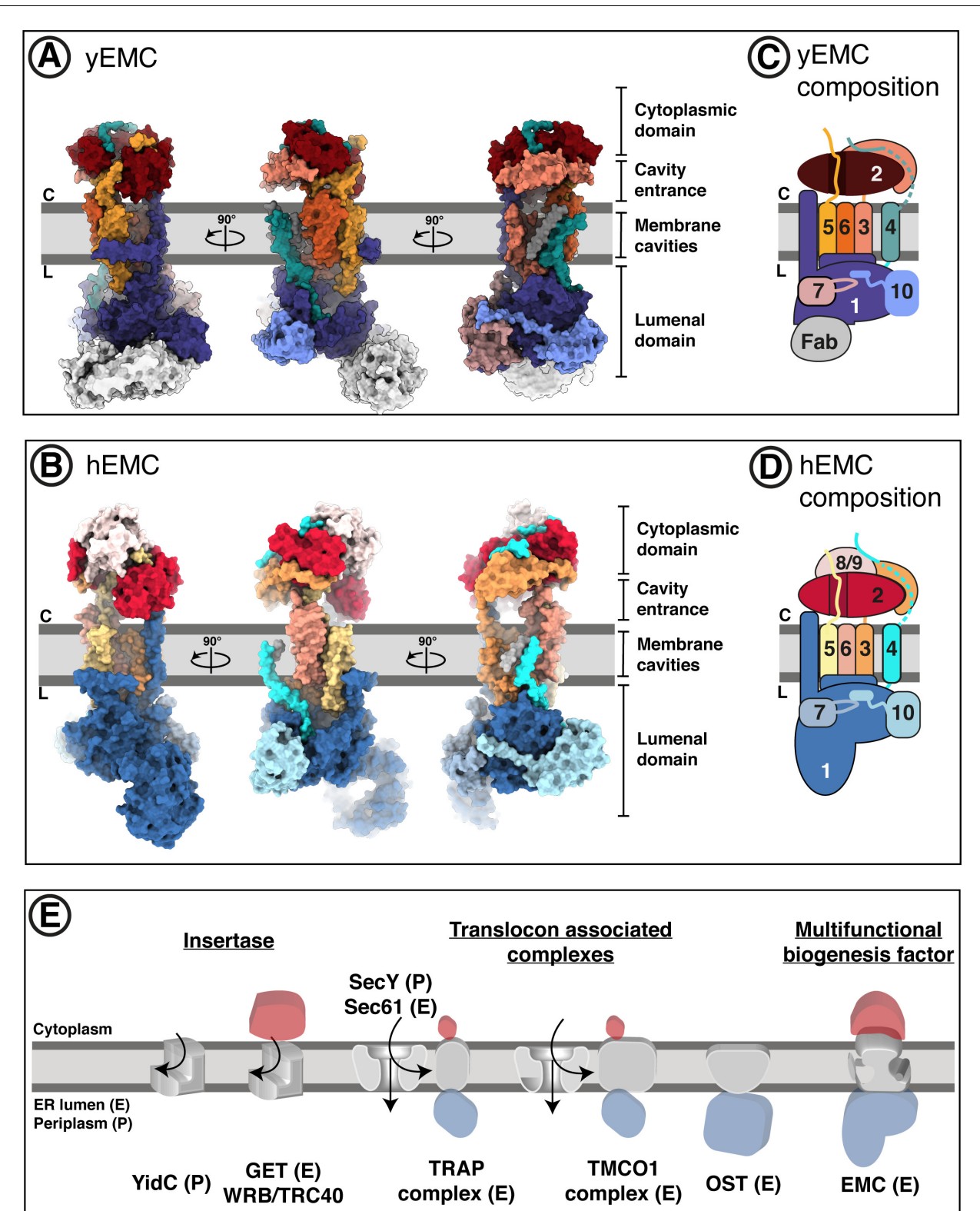

**Figure 2.** Overall structures of yeast and human EMC. (A) Cryo-EM structure of yEMC in nanodiscs. Three orthogonal views of the yEMC cryo-EM structure shown as surface rendering. Gray bars delineate the approximate ER membrane boundaries with the cytoplasmic (C) and lumenal (L) sides indicated. The FAb molecule bound to the yEMC1 lumenal domain is colored in gray. (B) Cryo-EM structure of hEMC in nanodiscs. Labeling as in (A). (C) Subunit composition and color scheme of yEMC used throughout the manuscript. Dotted line indicates a portion of yEMC4 unresolved in the cryo-

*Figure 2 continued on next page*

*Figure 2 continued*

EM map and left unmodeled. (D) Subunit composition and color scheme of hEMC used throughout the manuscript. (E) Schematic depiction and comparison of the EMC architecture to known transmembrane protein biogenesis factors in the ER and the bacterial plasma membrane. Cytoplasmic, transmembrane and lumenal domains are depicted as cartoons colored red, gray and blue, respectively. E, eukaryotic; P, prokaryotic.

The online version of this article includes the following figure supplement(s) for figure 2:

**Figure supplement 1.** Cryo-EM reconstruction of yEMC.
**Figure supplement 2.** Cryo-EM reconstruction of hEMC.
**Figure supplement 3.** Cryo-EM data processing workflow for yEMC.
**Figure supplement 4.** Cryo-EM data processing workflow for hEMC.
**Figure supplement 5.** yEMC cryo-EM map validation.
**Figure supplement 6.** hEMC cryo-EM map validation.
**Figure supplement 7.** Subunit-subunit correspondence between yEMC and hEMC.
**Figure supplement 8.** Comparison between individual yEMC and hEMC subunits.
**Figure supplement 9.** Pairwise superposition of EMC structures in the PDB.

The multi-protein cytoplasmic cap has distinct elements between hEMC and yEMC. Capping the cytoplasmic domain in hEMC is hEMC8/9; the functional roles of this cap-like structure are not yet clear. An hEMC8-9 heterodimer is not observed and our cryo-EM permits tracing with both the hEMC8 or hEMC9 amino acid sequence (*Figure 3—figure supplement 3*). Mass spectrometric analysis of our hEMC preparations reveals slightly higher abundance of hEMC8 to hEMC9 (*Figure 1—figure supplement 2E–F*, *Supplementary file 1*), so we modeled the cytoplasmic cap structure with the hEMC8 sequence. A groove on hEMC8/9 cradles an N-terminal peptide of hEMC4, which proceeds into the EMC4 segment that traverses over hEMC2 and the three-helix bundle of hEMC3 (*Figure 3D*). Although yEMC lacks EMC8/9, yEMC4 follows a similar binding trajectory along cytoplasmic yEMC2 and yEMC3 surfaces. A stretch of 20 hEMC4 amino acids (residues 23–42) after the hEMC8/9 binding site is only poorly resolved in our cryo-EM maps and predicted to be disordered (40% glycine content). This loop contains primarily polar amino acids, and traverses the top of the hEMC2 TPR domain. To see whether this dynamic hEMC2-hEMC4 interface played a role in client stabilization, we mutated two charged patches on hEMC2 to alanines ($hEMC2^{E146A+E149A+Q150A}$, $hEMC2^{E168A+D170A+K173A}$), lying in close vicinity to $hEMC4^{23-42}$ (*Figure 3E–F*). These mutants lead to a modest accumulation of the tail-anchored client ($SQS^{378-410}$) but did not affect polytopic client abundance or decrease of hEMC subunits (*Figure 3E–F*; *Figure 1—figure supplement 5*). Several mutants across the cytoplasmic domain showed similar phenotypes, supporting a key role in tail anchor protein biogenesis (*Figure 3—figure supplements 1–2*).

## Two distinct cavities are present in the transmembrane domain

The transmembrane core of EMC is predicted to include contributions from each subunit except for EMC2 and, in humans, hEMC8/9 (*Figure 2C–D*). The EMC presents two distinct and structurally conserved cavities on opposite sides of the transmembrane core that differ in size, shape, subunit compositions and apparent function (*Figure 4A–B*). One cavity, which we refer to as the lipid-filled cavity, appears contiguous with the ER lipid environment (*Figure 4A*). The second cavity, which we refer to as the gated cavity, appears to open toward the cytoplasm in our structures and is more occluded by a transmembrane helix gate from the lipid environment (*Figure 4B*). Notable structural hallmarks present in both species include a superimposable core of nine transmembrane helices, a set of flexible gate helices, and an amphipathic EMC1 brace helix (*Figure 4C*).

## The gated cavity serves as a conduit for terminal helix insertion

Evaluating potential client paths from the cytoplasm into the transmembrane domain revealed a cavernous opening at the membrane-cytoplasmic interface of the gated cavity, wide enough to allow passage of a client helix, and tapering toward the lumen (*Figure 4D*). Consistent with its potential role as a cytoplasmic conduit into the EMC, the EMC3 portion of the cytoplasmic domain, which delineates this opening, sits approximately 45 Å from the lumenal side of the gated cavity. This dimension exceeds the thickness of the ER membrane (*Mitra et al., 2004*; *Heberle et al., 2020*; *Cornell et al., 2020*; *Figure 4D*). This cavity is lined primarily by EMC3, EMC4, and EMC6 (*Figure 5A*). Simulating the dimension of the first transmembrane helix of a known terminal

**Table 1.** Cryo-EM data acquisition, reconstruction, and model refinement statistics.

| | yEMC in detergent dataset 1 | yEMC in detergent dataset 2 | yEMC in nanodiscs | hEMC in detergent | hEMC in nanodiscs |
|---|---|---|---|---|---|
| EMDB accession code | EMD-23033 | | EMD-23003 | EMD-11733 | EMD-11732 |
| PDB accession code | PDB-7KTX | | PDB-7KRA | PDB-7ADP | PDB-7ADO |
| **Data collection and processing** | | | | | |
| Microscope | FEI Technai Polara | FEI Titan Krios | FEI Titan Krios | FEI Titan Krios | FEI Titan Krios |
| Camera | Gatan K2 Summit | Gatan K2 Summit | Gatan K3 | Gatan K3 | Gatan K3 |
| Magnification | 31,000x | 22,500x | 105,000x | 81,000x | 105,000x |
| Voltage (kV) | 300 | 300 | 300 | 300 | 300 |
| Electron exposure (e⁻/Å²) | 56.8 | 58.3 | 67 | 62 | 72 |
| Defocus range (μm) | −1.0 to −3.0 | −1.0 to −3.0 | −0.8 to −2.5 | 0.7–2.8 | 0.7–2.8 |
| Pixel size (Å) | 1.22 | 1.31 | 0.853 | 1.094 | 0.8512 |
| Software | Relion 2.0, Relion 3.0, THUNDER | Relion 2.0, Relion 3.0, THUNDER | Relion 3.0, cryoSPARC v2 | Relion 3.0, cryoSPARC v2 | Relion 3.0, cryoSPARC v2 |
| Symmetry imposed | C1 | C1 | C1 | C1 | C1 |
| Initial particle images (no.) | 419,907 | 670,078 | 6,100,000 | 3,350,000 | 5,900,000 |
| Final particle images (no.) | 83,599 | 170,186 | 230,528 | 144,222 | 177,560 |
| Overall map resolution (Å) | 8 | 7 | | | |
| FSC threshold 0.143 | 4.3 (combined) | | 3.2 | 3.60 | 3.39 |
| Local map resolution range (Å) | 3.6–6.4 | | 2.6–6.4 | 2.8–6.0 | 3.0–7.2 |
| **Refinement** | | | | | |
| Software | Phenix 1.18 real-space-refine | | Phenix 1.18 real-space-refine | Phenix 1.18 real-space-refine | Phenix 1.18 real-space-refine |
| Model resolution (Å) | | | | | |
| FSC threshold 0.5 | 4.5 | | 3.5 | 3.9 | 3.6 |
| Map sharpening B factor (Å²) | −125 | | −75 | −115 | −126 |
| **Model composition** | | | | | |
| Non-hydrogen atoms | 17,315 | | 17,293 | 15,040 | 16,652 |
| Protein residues | 2171 | | 2164 | 1880 | 2086 |
| Ligands | NAG: 6 | | NAG: 5; PCW: 1 | NAG: 2 | NAG: 4; PCW: 5 |
| **B factors (Å²)** | | | | | |
| Protein (mean) | 167 | | 107 | 111 | 126 |
| Ligand (mean) | 146 | | 92 | 107 | 127 |
| **R.m.s. deviations** | | | | | |
| Bond lengths (Å) | 0.006 | | 0.005 | 0.005 | 0.005 |
| Bond angles (°) | 0.988 | | 0.804 | 0.718 | 0.782 |
| **Validation** | | | | | |
| MolProbity score | 1.44 | | 1.29 | 1.46 | 1.28 |
| Clashscore | 8.1 | | 3.5 | 9 | 6 |
| **Ramachandran plot** | | | | | |
| Favored (%) | 98 | | 97 | 97 | 97 |

*Table 1 continued on next page*

*Table 1 continued*

| | yEMC in detergent dataset 1 | yEMC in detergent dataset 2 | yEMC in nanodiscs | hEMC in detergent | hEMC in nanodiscs |
|---|---|---|---|---|---|
| Allowed (%) | 2 | | 3 | 3 | 3 |
| Disallowed (%) | 0 | | 0 | 0 | 0 |

insertase-client (B1AR - *Chitwood et al., 2018*) suggests that there is sufficient space for a client helix even in the client-free state of the EMC (*Figure 5B*). The gated cavity is hydrophilic on the cytoplasmic side and becomes increasingly hydrophobic toward the lumenal side (*Figure 5C*).

The entrance into the gated cavity interior (*Figure 5A*) is formed primarily by the EMC3 cytoplasmic domain. To test its function, charge swap mutations were introduced along the rim of this opening (hEMC3$^{E63K + D213K + E223K}$, hEMC3$^{R59E + R62E + K216E}$) (*Figure 5D*). These mutants resulted in loss of the tail-anchored client (SQS$^{378-410}$) and partial loss of the N-terminal insertase-dependent polytopic client (B1AR), reflecting a failure to support insertase activity. These mutants had no appreciable effect on the abundance of the polytopic transmembrane client (TMEM97) reporter (*Figure 5E*, *Figure 5—figure supplements 1–2*). A similar phenotype was observed with alanine substitutions for a pair of lysines at the periphery of this cytoplasmic rim (hEMC3$^{K42A + K43A}$) (*Figure 5—figure supplement 2*).

Having identified a functionally important entry route for terminal helix insertase clients, we next considered potential surfaces inside the cavity that might accommodate a client helix. A polar patch close to the membrane interior of this cavity was conspicuous, even though the specific amino acid residues are not strictly conserved (*Figure 1—figure supplement 8*). Mutating a pair of adjacent asparagine residues to equivalently sized but negatively charged aspartates (hEMC3$^{N114D+N117D}$) resulted in a dramatic decrease in SQS$^{378-410}$ reporter levels and no significant decrease in the other two client reporter levels (*Figure 5F*). Western blot analysis for this mutant showed wild-type rescue levels of hEMC subunits and a decrease in endogenous SQS levels (*Figure 1—figure supplement 6*). Meanwhile, mutating a neighboring positively charged residue to an alanine (hEMC3$^{R180A}$), a residue that is conserved in some of the YidC-superfamily insertase proteins (*Anghel et al., 2017*), resulted in partial loss of only the tail-anchored insertase client (SQS$^{378-410}$) (*Figure 5—figure supplements 1* and *3*).

Lastly, we surveyed residues closer to the hydrophobic lumenal side of the gated cavity. Lipid density was resolved at positions along the cavity in hEMC and yEMC cryo-EM maps (*Figure 4B*) and the properties of this hydrophobic seal to the lumen are conserved (*Figure 5—figure supplement 4A–B*). The importance of this hydrophobic seal is suggested by the strong effect of a structurally mild mutation of a conserved methionine to a leucine (hEMC3$^{M151L}$), which caused significant decrease in both SQS$^{378-410}$ and B1AR abundance (*Figure 5G*). Mutation of a neighboring aromatic residue (hEMC3$^{F148L}$), contacting both a lipid and a hEMC4 C-terminal transmembrane helix, caused a decrease in all three client types without altering the levels of hEMC subunits (*Figure 5—figure supplements 1–2*, *Figure 1—figure supplement 6*). Together these results indicate that proper EMC insertase function depends on the exact composition of the cavity and not simply on its hydrophobic nature.

## Structural heterogeneity suggests a role for the gate in regulating access to the insertase transmembrane cavity

While the core transmembrane helices of the gated cavity are superimposable in all four of our EMC structures, the adjacent gate helices appear in different relative orientations. The structural variability likely reflects dynamics of the gate (*Figure 4C*). Comparing detergent and nanodisc maps for both species identified two major gate conformations (*Video 2*, *Figure 5—figure supplement 5A*). One of the conformations, referred to as the closed-gate conformation, results in a more occluded membrane cavity. The other conformation, referred to as the open-gate conformation, would provide space for client accommodation.

The C-terminal transmembrane helix of EMC4 and ensuing lumenal segment are well resolved in all four structures; however, other regions of EMC4, including the segment connecting the

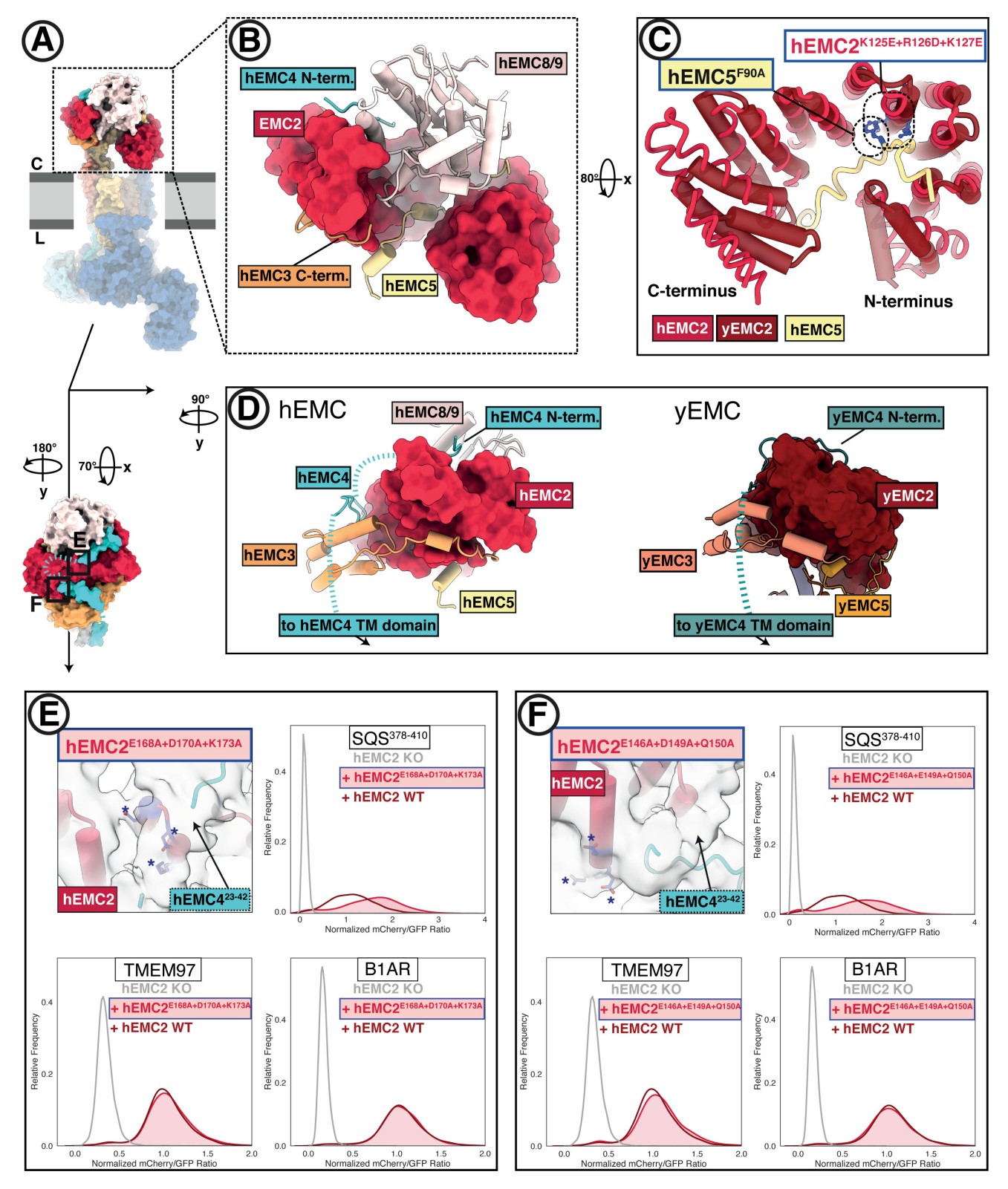

**Figure 3.** The EMC cytoplasmic domain contains conserved functional interfaces and may engage C-tail-anchored clients directly. (**A**) Position of the hEMC cytoplasmic domain relative to the membrane and the rest of the complex. Shown is the surface rendered hEMC structure reconstituted in nanodiscs. (**B**) EMC2 nucleates a protein-protein interaction hub in the cytoplasm. Zoomed-in view of the cytoplasmic domain from (**A**). EMC2 is shown as surface rendering while interacting EMC subunits are shown as cartoon cylinders. (**C**) EMC2 forms a TPR domain which binds EMC5. Overlaid are

*Figure 3 continued on next page*

Figure 3 continued

hEMC2 (red) and yEMC2 (dark red), illustrating the more tightly wound yEMC2 TPR solenoid. Two mutants, one in EMC5 and three in EMC2, are colored in blue, and show destabilizing phenotypes for EMC integrity. (D) A cytoplasmic cap structure involving EMC4 is conserved in yEMC and hEMC. Shown is a side-by-side comparison between the cytoplasmic domains of hEMC (left) and yEMC (right), highlighting the similar path EMC4 takes from the cytoplasmic domain toward the transmembrane domain. While an interaction surface between EMC8/9 and the EMC4 N-terminus is absent in yeast, yEMC4 binds at the top of the EMC2 TPR domain and assumes as similar position across the EMC3 cytoplasmic domain at the cytoplasm-membrane interface. (E) Fluorescent client reporter stability assay for TMEM97 (N-cytoplasmic polytopic client), B1AR (N-lumenal polytopic client) and $SQS^{378-410}$ (C- lumenal tail-anchored client) in EMC2 KO cells expressing mutant $hEMC2^{E168A+D170A+K173A}$ (shaded) or WT hEMC2 rescue (unshaded). Shown is the model of hEMC in nanodiscs superposed with the unsharpened cryo-EM map, where the weaker density for EMC4 (23–42) becomes apparent. Mutated residues are colored blue and marked with asterisks for clarity. (F) Fluorescent client reporter stability assay, as in E, for the $hEMC2^{E146A+E149A+Q150A}$ mutant.

The online version of this article includes the following figure supplement(s) for figure 3:

**Figure supplement 1.** Flow cytometry for mutations in the EMC cytoplasmic domain.
**Figure supplement 2.** Additional flow cytometry for mutations in the EMC cytoplasmic domain.
**Figure supplement 3.** Both EMC8 and EMC9 can be fitted into the hEMC cryo-EM maps.

cytoplasmic domain to the transmembrane gate helices, were poorly resolved, perhaps owing to mobility. The yEMC detergent map, yEMC nanodisc map, and hEMC detergent map all show the unassigned helices in the closed conformation, preventing client residence in the gated cavity. By contrast, the hEMC nanodisc map reveals an open-gate conformation with the unassigned helices shifted away from the transmembrane core to provide space for a client (*Figure 5B*). Consistent with our observations, the closed transmembrane gate conformation can also be seen in recently published cryo-EM maps of hEMC (*O'Donnell et al., 2020*) and yEMC (*Bai et al., 2020*), which studied LMNG and digitonin-solubilized complexes, respectively (*Figure 5—figure supplement 5B*). We note that the conformational heterogeneity and concomitant lower resolution of the gate likely accounts for the challenges in making unambiguous subunit assignments (*Figure 5—figure supplement 5C–E*), reflected by the three different interpretations reported in recent structures (*Pleiner et al., 2020*; *O'Donnell et al., 2020*; *Bai et al., 2020*).

Considering the apparent flexibility of the gate, we sought to mutate the hEMC4 interfaces resolved in the cytoplasm versus the membrane. As described above, mutating residues that together form a composite-binding surface for the cytoplasmic domain of hEMC4 ($hEMC2^{E146A+E149A+Q150A}$, $hEMC2^{E168A + D170A + K173A}$, *Figure 3E–F*), we observed a modest accumulation of the tail-anchored insertase client ($SQS^{378-410}$). Likewise, mutating residues in the center of the gated cavity, close to one of the unassigned helices in the closed-gate conformation ($hEMC3^{V118A + I122A}$) (*Video 2*, *Figure 5—figure supplement 1*) led to an increase of $SQS^{378-410}$. This $SQS^{378-410}$ accumulation effect stands in contrast to mutating a residue that contacts the lumenal anchor of hEMC4 ($hEMC3^{F148L}$), which caused a reduction of $SQS^{378-410}$ levels (*Figure 5—figure supplement 1*).

## The lipid-filled cavity is critical for both insertase-dependent and insertase-independent EMC functions

In addition to the gated cavity, the EMC harbors another membrane-accessible cavity. The surface of the lipid-filled cavity includes contributions from EMC1, EMC3, EMC5, and EMC6 (*Figure 6A*). In our structures, the EMC2 N-terminus occludes cytoplasmic accessibility to this cavity (*Figure 4D*, *Figure 6A–B*). However, this cavity may be accessible from the membrane or the ER lumen. The respective distance from the cytoplasmic EMC2 N-terminus to the lumenal side of the lipid-filled cavity is approximately 35 Å across, which is close to the average ER membrane thickness (*Mitra et al., 2004*).

The lipid-filled cavity features a uniformly hydrophobic surface (*Figure 6C*) and superimposes across our ensemble of EMC structures. As noted, we resolved several lipids in our cryo-EM maps lining the cavity wall and modeled four POPC (1-palmitoyl-2-oleoyl-sn-glycero-3 phosphatidylcholine) molecules in the hEMC nanodisc map (*Figure 6C*). The residues in close proximity to these lipids are moderately conserved (*Figure 5—figure supplement 4C–D*). To characterize the functional role of the lipid-filled cavity, we mutated cavity-lining and lipid-proximal residues (*Figure 6D*, *Figure 6—figure supplements 1–2*). Most of these mutations resulted in an increased abundance of the tail-anchored reporter ($SQS^{378-410}$) and wild-type rescue levels for the other two reporters (B1AR,

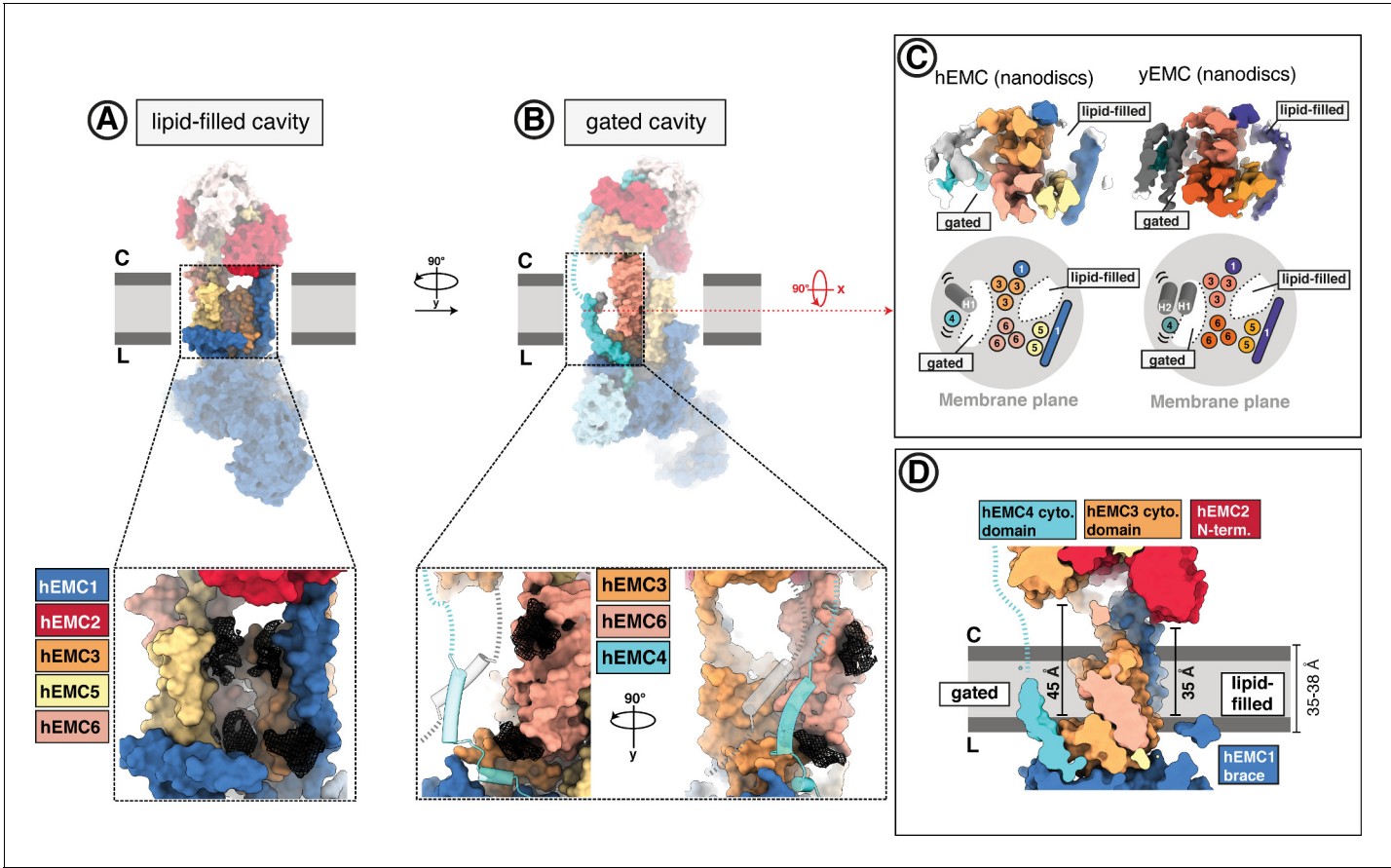

**Figure 4.** The EMC houses two transmembrane cavities with conserved core structures and distinct accessibilities. (**A**) Location and composition of the lipid-filled cavity. A zoom-in view on the cavity is shown below, which is composed of EMC1, EMC3, EMC5, and EMC6. Resolved lipid densities from the cryo-EM map of hEMC in POPC nanodiscs are shown as black mesh zoned within 3 Å of modeled POPC molecules. (**B**) Location and composition of the gated cavity. Two orthogonal zoom-in views of the cavity are shown below, which is composed of EMC3 and EMC6. A transmembrane gate opposite the cavity wall is depicted as transparent cartoon cylinders and has contributions from the C-terminal EMC4 transmembrane helix along with up to two additional, unassigned helices. Resolved lipid densities are shown as in (**A**). (**C**) The dual-cavity architecture of the EMC transmembrane domain is conserved between yEMC and hEMC. Unsharpened cryo-EM maps of hEMC and yEMC in nanodiscs (top) are shown along with corresponding schematic representations of the spatial organization of all transmembrane helices (bottom). The gate helices of the gated cavity represent the region of highest conformational heterogeneity across our collection of EMC structures. (**D**) The two EMC transmembrane cavities feature distinct accessibilities. Shown is a central slice through the surface rendered hEMC nanodisc structure with the two membrane cavities on opposite sides. Measuring from the lumenal to the cytoplasmic side, gated and lipid-filled cavities measure 45 Å and 35 Å across, respectively. This suggests that the gated cavity has accessibility from the cytoplasm while the lipid-filled cavity does not.

TMEM97). However, one lipid-proximal mutant showed decreased levels of all three client reporter types with varying severity (hEMC3$^{R13E}$) without altering overall EMC levels (*Figure 6E*). Western blotting for the endogenous SQS and TMEM97 revealed a decrease in endogenous SQS and TMEM97 levels for this mutant (*Figure 1—figure supplement 6*). An analogous mutation in *Drosophila* EMC3 was recently was reported to cause reduced levels of Rh1 in this mutant background (*Xiong et al., 2020*). The amphipathic EMC1 brace helix, which packs against the transmembrane helices of EMC5, is a structural hallmark of the lipid-filled cavity (*Figure 6D*). Here, mutating interfacial residues from hEMC5 (hEMC5$^{H19L+S23A+Q26L}$) caused a marked decrease in the N-lumenal polytopic reporter (B1AR) and no effect on either the tail-anchored client (SQS$^{378-410}$) or the polytopic client reporter (TMEM97) (*Figure 6F*). Unexpectedly, mutating interfacial residues from hEMC1 (hEMC1$^{F473Y+R487K}$) showed a diametrically opposed phenotype in which B1AR was unaffected, increased SQS$^{378-410}$ levels, and TMEM97 levels markedly decreased (*Figure 6G*). Another mutation in this brace (hEMC1$^{M483A+R487H+Q491N}$) resulted in a decrease in TMEM97 and no significant effect on the other two client reporters. An adjacent hEMC5$^{D44K}$ mutations in the interfacial brace had yet

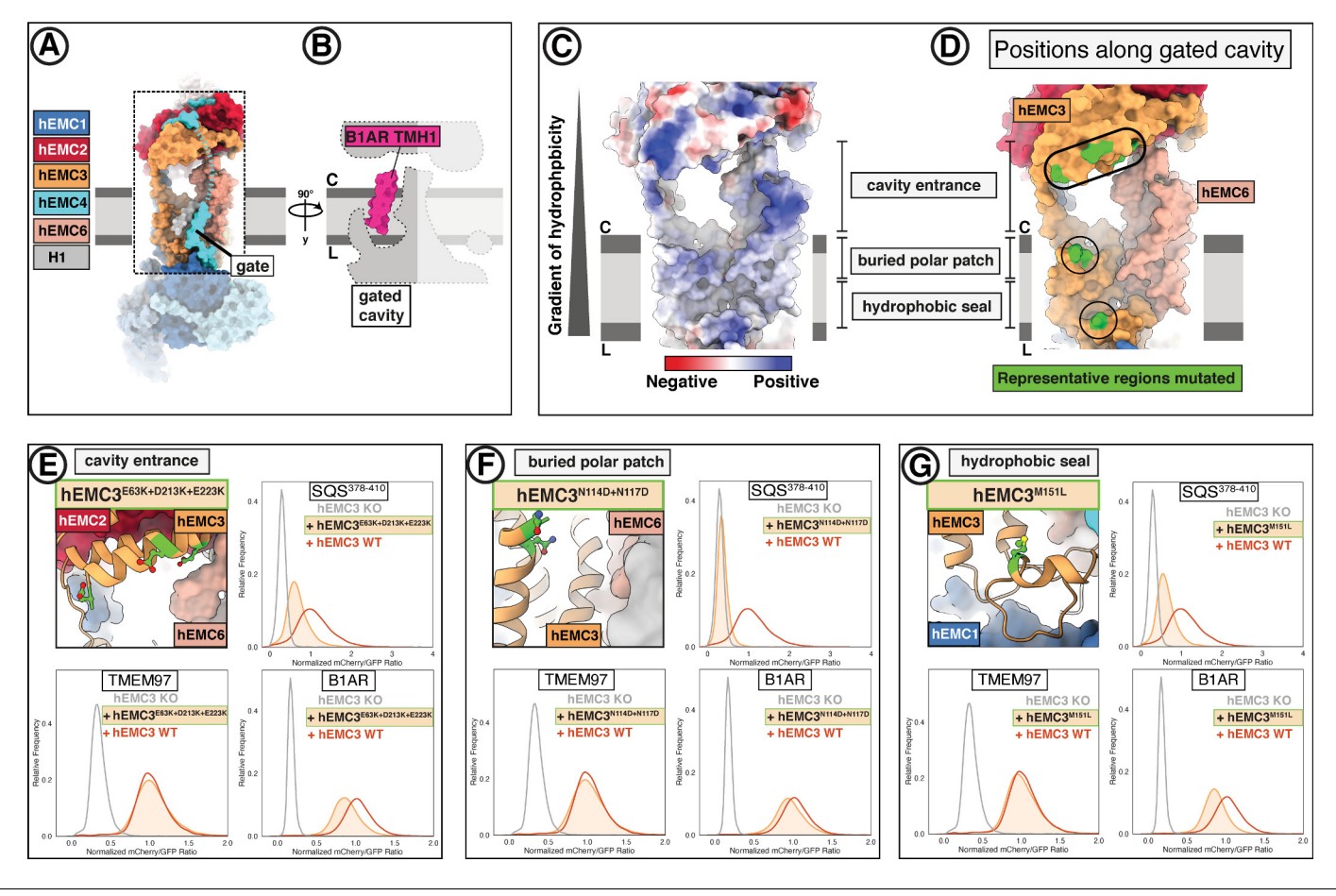

**Figure 5.** EMC houses an insertase module centered on EMC3 in the gated membrane cavity. (**A**) A transmembrane gate anchored in the cytosol and the lumen is a structural hallmark of the EMC gated cavity. Shown is a surface rendering of the hEMC model in lipid nanodiscs with an unresolved EMC4 connection between the cytoplasm and the membrane depicted as a dashed line. An unassigned helix of the gate is shown in gray (H1). (**B**) The gated cavity in the hEMC nanodisc structure has sufficient space to accommodate a client transmembrane helix. The space-filling model of the first transmembrane helix of B1AR (B1AR TMH1) is shown placed inside an outline of the EMC gated cavity. (**C**) A hydrophobic gradient characterizes the surface of the EMC gated cavity from the cytoplasmic to the lumenal side. Gate helices have been omitted for clarity. The surface of the hEMC nanodisc structure is colored by electrostatic surface potential ranging from −15 (red) to +15 (blue) kcal/(mol·e). (**D**) Distinct EMC3 regions along the gated cavity hydrophobic gradient targeted for mutagenesis. Mutated residues are colored in lime. (**E**) Fluorescent client reporter stability assay for the EMC3 cavity entrance mutant, hEMC3$^{E63K+D213K+E223K}$. (**F**) As in (**E**) for the EMC3 buried polar patch mutant, hEMC3$^{N114D+N117D}$. (**G**) As in (**E**) for the EMC3 hydrophobic seal mutant, hEMC3$^{M151L}$.

The online version of this article includes the following figure supplement(s) for figure 5:

**Figure supplement 1.** Flow cytometry of gated cavity mutants.

**Figure supplement 2.** Additional flow cytometry of gated cavity mutants.

**Figure supplement 3.** Comparison of EMC3 to YidC-family members.

**Figure supplement 4.** Resolved lipid densities in hEMC and yEMC nanodisc maps.

**Figure supplement 5.** Comparison of gate conformations.

different resulting client flow cytometry profiles, with an increase in SQS$^{378-410}$ and no effect on either of the polytopic client reporters (*Figure 6—figure supplement 2B–C*). The pleiotropic client phenotypes across the panel of interfacial brace mutants suggest that this feature is critical for multiple EMC functions.

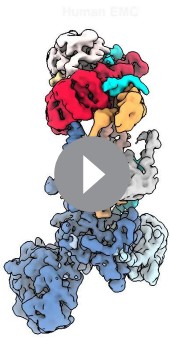

**Video 2.** EMC transmembrane cavity gate conformations. Overview of hEMC colored and labeled by subunit. Volume fades away to hEMC nanodisc model. hEMC nanodisc model remains constant as segmented maps of the unassigned gate helices are shown of hEMC detergent, yEMC detergent, and yEMC nanodisc maps. hEMC is colored cyan, yEMC is colored dark cyan, and gate helices are colored in shades of gray and purple as indicated by the label on the left. Two residues are shown in stick representation colored gold.

https://elifesciences.org/articles/62611#video2

## The EMC lumenal domain is crucial for the biogenesis of multi-pass transmembrane proteins

Composed primarily of EMC1, EMC7, and EMC10, the extensive EMC lumenal domain (*Figure 7A*) is important for polytopic client biogenesis and interactions with lumenal chaperones (*Luo et al., 2002*; *Shurtleff et al., 2018*; *Hiramatsu et al., 2019*; *Coelho et al., 2019*). EMC7 and EMC10 are scaffolded on two beta-propellers of EMC1, one distal and the other proximal to the membrane. The lumenal cap differs between hEMC and yEMC, with a four-bladed distal beta-propeller in yeast and eight-bladed distal propeller the human complex (*Figure 7B*). All three lumenal EMC subunits have structural folds known to participate in protein-protein interactions (*Reinisch and De Camilli, 2016*). Mutations in this lumenal domain have been linked to loss of the EMC complex (*Bircham et al., 2011*), a trafficking delay for membrane protein Pma1 (*Luo et al., 2002*), and male infertility (*Zhou et al., 2018*).

Several regions of the lumenal domain form stabilizing interactions with the membrane cavities. The gate helices of the gated cavity are anchored via the embedding of EMC4's C-terminus within the membrane-proximal EMC1 propeller. The lipid-filled cavity is connected to the ER lumenal domain via the amphipathic EMC1 brace helix, which is tethered to the membrane-proximal EMC1 beta-propeller. The connections between the lumenal domain and the transmembrane cavities could allow for conformational coupling during client handling. Indeed, superimposing the two conformations presented above, the open- and closed-gate states, revealed not only differences in the transmembrane domain but also a rotation of the lumenal domain relative to the membrane cavities (*Video 3*). The lumenal positioning is consistent for all three of our closed-gate conformation reconstructions (hEMC detergent, yEMC nanodisc, yEMC detergent). By contrast, the one map with an open gated cavity displayed a lumenal rotation and concomitant shifts in position of the hEMC1 brace helix (*Figure 7—figure supplement 1*). Indeed, our set of interfacial hEMC1 brace mutants described above (*Figure 6F–G*, *Figure 6—figure supplement 2B–C*), showed differing client phenotypes when mutated from either the hEMC1 or the hEMC5 side. This suggests a complex conformational interplay between lumenal and transmembrane domains during the engagement of diverse client types.

We investigated several known disease mutations in both conserved and human-specific regions of hEMC1 (*Figure 7C–D*, *Figure 7—figure supplements 2–3*; *Harel et al., 2016*; *Abu-Safieh et al., 2013*; *Amberger et al., 2019*). One of these disease-associated residues sits near the anchor point for the lumenal hEMC4 transmembrane gate helix (hEMC1$^{R881C}$), while the majority are found farther from the membrane (hEMC1$^{G868R}$, hEMC1$^{A144T}$, hEMC1$^{T82M}$) (*Figure 7C–D*, *Figure 7—figure supplement 2B*). Incorporating each of these disease mutations into our EMC functional assay resulted in lower levels of the N-cytoplasmic polytopic client (TMEM97) and an increase in the level of the tail-anchored client (SQS$^{378-410}$), discussed in more detail below.

Two different hEMC1 mutants associated with cerebellar atrophy, visual impairment, and psychomotor retardation (hEMC1$^{T82M}$, hEMC1$^{G868R}$) map to the hinge region between the hEMC1 beta propellers where hEMC7 binds (*Figure 7D*). Both the mutants at this protein-protein interface resulted in depletion of the N-cytoplasmic polytopic client (TMEM97). EMC7 and EMC10 form beta-sandwich domains on either side of the membrane-proximal beta-propeller of EMC1 and contact each other across the EMC1 surface. Consistent with our structures, coupling of these subunits is supported by the prior finding that in the absence of EMC7, EMC10 is also lost from the complex

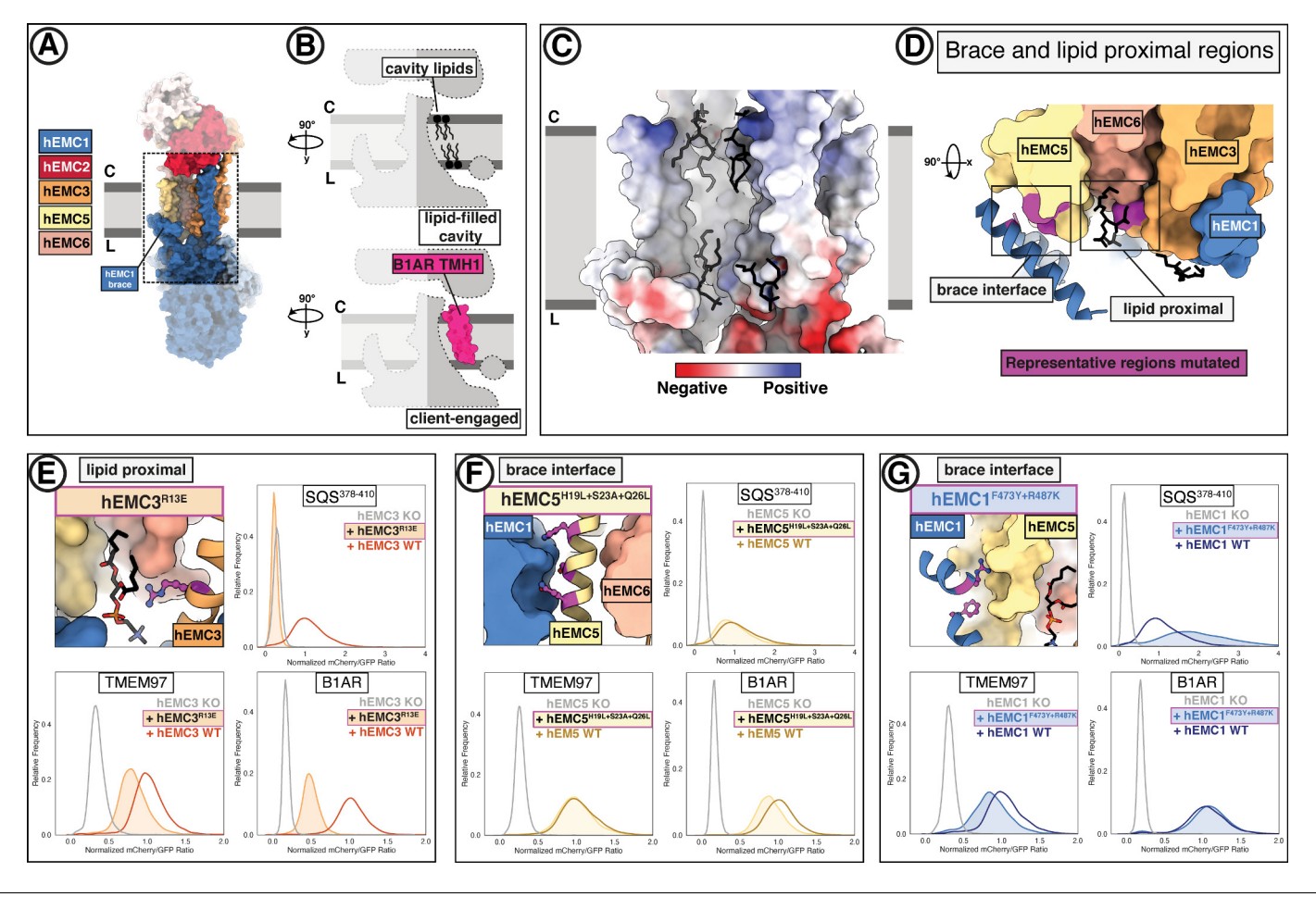

**Figure 6.** A lipid-filled cavity in the EMC transmembrane domain stabilizes disparate client proteins. (**A**) An EMC1 amphipathic brace helix delineates the boundary of the lipid-filled transmembrane cavity and packs against EMC5. Shown is a surface rendering of the hEMC model in nanodiscs. EMC4, EMC5, EMC6, and EMC1 subunits all contribute to the cavity lining. (**B**) The lipid-filled cavity in the hEMC nanodisc is occupied by several lipid molecules. Cartoon outlines of the gated cavity illustrate that the cavity could in principle allow for occupancy of a client helix (B1AR TMH1), possibly by lipid displacement or movement of the EMC1 brace helix. (**C**) The lipid-filled cavity has a uniform hydrophobic lining. Shown is an electrostatic surface rendering of the hEMC nanodisc structure colored as in *Figure 5C*. The cytoplasm-membrane interface contains positively charged residues and the lumenal interface contains negatively charged residues. Modeled phospholipid molecules are displayed in black. (**D**) Lipid-proximal and brace interface residues targeted for mutagenesis. Selected regions targeted for mutagenesis are colored in magenta and include brace interface mutations both in EMC1 and EMC5, as well as a lipid-proximal residue in EMC3. (**E**) Fluorescent client reporter stability assay for the hEMC3$^{R13E}$ mutant, which is in close proximity to a modeled POPC molecule. (**F**) As in (**E**) for the hEMC5$^{H19L+S23A+Q26L}$ mutant, which sits at the interface to the EMC1 amphipathic brace helix. (**G**) As in (**E**) for the hEMC1$^{F473Y+R487K}$ mutant, which sits at the interface to the EMC5 transmembrane helices.

The online version of this article includes the following figure supplement(s) for figure 6:

**Figure supplement 1.** Flow cytometry of lipid-filled cavity mutants.

**Figure supplement 2.** Additional flow cytometry of lipid-filled cavity mutants.

while the other EMC components appear unaffected (*Shurtleff et al., 2018*). EMC7 and EMC10 have been proposed to be auxiliary components with weaker phenotypes compared to core EMC subunits (*Jonikas et al., 2009*; *Shurtleff et al., 2018*; *Dickinson et al., 2016*). Upon deleting yEMC7, multi-pass transmembrane clients are retained in the ER but tail-anchored clients, including SQS-homolog Erg9, decrease in abundance (*Shurtleff et al., 2018*).

Several features of our data suggest dynamic association of hEMC7. Density for the hEMC7 beta-sandwich at the hinge between the two hEMC1 beta propellers was relatively weak in the consensus hEMC nanodisc map (*Figure 2—figure supplement 4*). Additional rounds of 3D classification revealed two distinct classes, one with clear density for hEMC7, and one with weak density in this

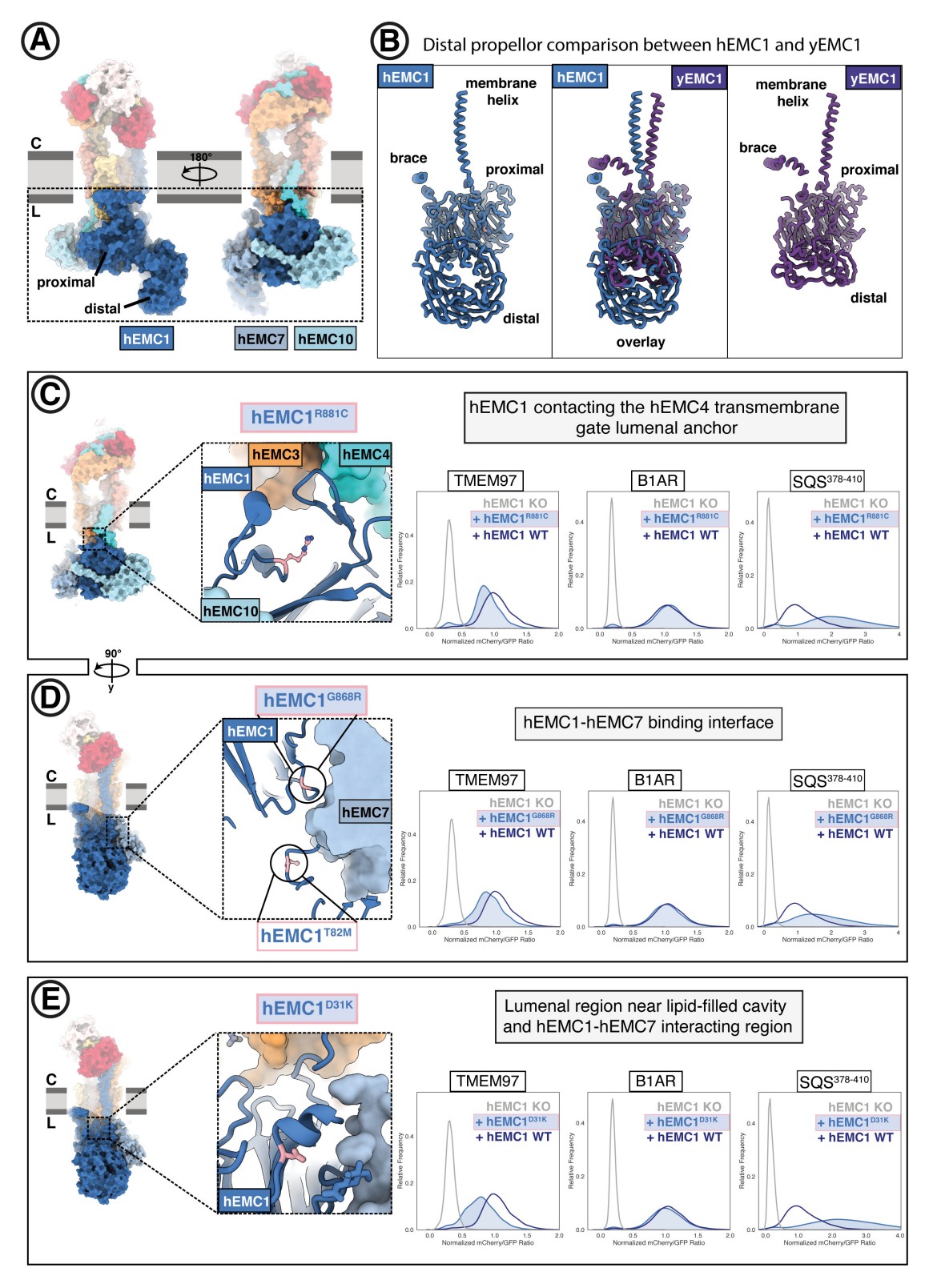

**Figure 7.** The large EMC lumenal domain is the site for several annotated disease mutations. (**A**) Two views of the hEMC nanodisc structure. Two beta propellers are present in EMC1, one proximal to the membrane and one distal. (**B**) EMC1 is the largest EMC subunit and differs in size between yeast and human. Shown are human EMC1 (nanodisc), an overlay of human and yeast EMC1 (both nanodisc), and yeast EMC1 (nanodisc). (**C**) The

*Figure 7 continued on next page*

*Figure 7 continued*

hEMC1[R881C] mutant sits near the EMC4 lumenal gate anchor. Left: Location of the mutation (colored pink). Right: Fluorescent client R881C reporter stability assay for hEMC1. (D) As in (C) for for the hEMC1[G868R] mutant. (E) As in (C) for the hEMC1[D31K] mutant.

The online version of this article includes the following figure supplement(s) for figure 7:

**Figure supplement 1.** Conformational heterogeneity of the hEMC lumenal domain between detergent and nanodisc maps.
**Figure supplement 2.** Flow cytometry of lumenal domain mutants.
**Figure supplement 3.** Additional flow cytometry of lumenal domain mutants.

region. Mass spectrometric analysis of purified hEMC, however, revealed that the abundance of hEMC7 was similar to that of the other hEMC components (*Figure 1—figure supplement 2*; *Supplementary file 1*). Both reconstructions, with and without density for the hEMC7 lumenal domain, displayed well-resolved density for hEMC10. Together, we conclude that hEMC7 is associated with hEMC1 in two different conformational states of hEMC7 with potentially distinct functions.

The OMIM database (*Amberger et al., 2019*) lists a mutation of unknown significance linked to retinitis pigmentosa (hEMC1[A144T]) residing in the EMC1 distal propeller (*Figure 7—figure supplement 2*). Additionally, we also generated mutations in two surface exposed patches of the membrane-distal EMC1 beta-propeller projecting into the lumen (hEMC1[D31K], hEMC1[R69D], hEMC1[G71S], hEMC1[H93D + E138D + N282K], *Figure 7E*, *Figure 7—figure supplements 2–3*). Overall, these mutations displayed the same client effect: a decrease in the N-cytoplasmic polytopic client reporter (TMEM97), no change in the N-lumenal polytopic client reporter (B1AR), and accumulation of the tail-anchored client reporter (SQS[378-410]). Upon identifying antibodies against yEMC, we observed that the top two antibodies bind to a similar extended loop in the distal yEMC1 beta-propeller, perhaps suggesting that this site is accessible for co-factor binding in the ER. Intriguingly, this region of the lumenal domain corresponds to the region where hEMC1 has an expanded distal beta-propeller. Taken together, the data provide evidence that the lumenal domain is functionally coupled to the broader EMC role in transmembrane client stabilization. Moreover, these data support that the EMC is acting as a holdase chaperone to shield polytopic clients from degradation while they are folding to their functional form.

## Discussion

Our collection of yeast and human EMC structures revealed the intricate and dynamic architecture of this multifunctional transmembrane molecular machine. The structures served as the starting point for our systematic dissection of EMC's multifaceted functions by exploring the impact of structure-based mutations on the ability of the EMC to support the biogenesis of representative members of three classes of membrane proteins: SQS, a tail-anchored protein, which exploits EMC's C-terminal insertase activity; B1AR, which relies on EMC's N-terminal insertase activity; and TMEM97, a polytopic membrane protein, which depends on the EMC for its biogenesis but does not rely on either of EMC's terminal insertase activities. Our data revealed that a conserved dual membrane cavity architecture supports the biogenesis of this diverse panel of transmembrane clients.

Overall, our studies present a nuanced picture of EMC's multifunctionality, revealing structural regions that differentially impact production of the three distinct client types. Unexpectedly, we also find that alterations to either the cytoplasmic or lumenal domain of EMC lead to enhanced abundance of the TA

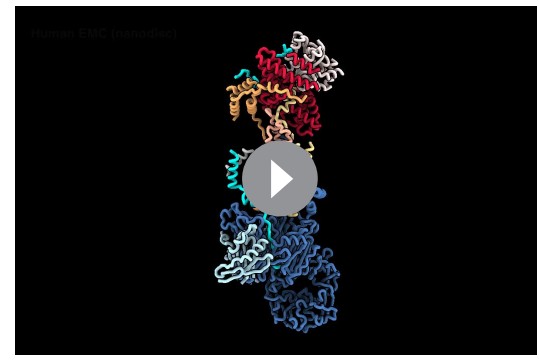

**Video 3.** hEMC lumenal domain differences between nanodisc and detergent models. Overview of hEMC nanodisc model colored and labeled by subunit. Structural landmarks are labeled. hEMC detergent model (colored gray) fades in and both models rotate. As the models rotate several structural features are highlighted.
https://elifesciences.org/articles/62611#video3

substrate. Moreover, our work provides a foundational framework for understanding how discrete yet allosterically coupled regions of the complex enable the multiple functions of the EMC to support membrane protein biogenesis. Taken together, these studies suggest a model in which the EMC differentially regulates the biogenesis of distinct membrane proteins, thereby contributing to cellular coordination of membrane protein abundance in accordance with physiological needs. We propose a model of the EMC functioning both as a terminal insertase as well as a holdase chaperone that is potentially modulated by post-translational modifications, lipid interactions, and protein-protein interactions (*Figure 8*). Here, we summarize our findings into a proposed model of EMC function for these three clients.

## Terminal insertase clients require an embedded insertase module within the EMC

EMC3's fold at the interface between the cytoplasm and membrane forms the core of the gated cavity and is reminiscent of proteins from the YidC family of insertases (*Borowska et al., 2015*; *Dalbey and Kuhn, 2015*; *Anghel et al., 2017*). Indeed, mutations in either the cytoplasmic or transmembrane domains of EMC3 establish that these features are critical for terminal helix insertase activity. In light of our observation of multiple gate conformations, we speculate that these conformations modulate insertion and release into the ER membrane.

Notably, mutating the surface of the cytoplasmic cap, which extends beyond the EMC3 cytoplasmic helices toward EMC8/9, resulted in an unexpected increase in C-tail anchor client (SQS[378-410]) abundance. Of the three clients analyzed, SQS was the only one to show enhanced levels. It is unclear if this enhancement is SQS-specific or representative more broadly of all post-translationally targeted EMC tail-anchored clients. Future studies will be required to address if this is due to regulated insertion of SQS by the EMC, parallel pathways for inserting SQS into the membrane (i.e. mediated by TRC40/GET), and/or slower cytoplasmic clearance of chaperone-bound SQS.

Post-translational insertase clients have previously been shown to be targeted to the ER by cytoplasmic chaperones (*Guna et al., 2018*). Structural analysis and coupled mutagenesis, from our and recent studies (*O'Donnell et al., 2020*; *Pleiner et al., 2020*; *Bai et al., 2020*), suggest that clients then engage the cytoplasmic domain of the EMC, the transmembrane gate opens, the terminal helix is inserted into the EMC-gated cavity, and then another conformational change would allow for release into the lipid bilayer (*Figure 8A–B*). Further studies are needed to establish the precise C-terminal client range, as most tail anchor clients have been shown to be inserted by the GET (in yeast) or WRB (in human) complexes (*Denic et al., 2013*; *Mateja and Keenan, 2018*).

## Both EMC cavities have resolved lipids and are critical for client biogenesis

Both the N-terminal (B1AR) and C-terminal insertase (SQS) clients depend on the EMC-gated cavity. Indeed, both the SQS tail-anchored helix and the first transmembrane helix of B1AR are moderately hydrophobic, with polar residues near the cytoplasmic end of the transmembrane helix, and both showed a strong dependence on the gated cavity. Nevertheless, our panel of mutants revealed some notable differences in the handling of these two client types. B1AR showed more dependence than SQS on the lipid-filled cavity in contrast to mutants elsewhere in the complex. Consistent with this, a number of mutations, primarily in the gated cavity, show residues of importance to both SQS and B1AR. However, there are also a number of mutations that appear to only affect SQS. One possible reason could be due to differences in the mechanism of initial engagement: SQS is targeted to the ER by cytoplasmic chaperones, while B1AR is targeted by SRP. Another key difference is that B1AR is polytopic and needs to overcome the additional challenge of tertiary transmembrane packing to reach its folded state. This work provides support for a model where the EMC inserts both types of terminal transmembrane helices into the gated cavity with differences in initial targeting and perhaps release into the lipid environment (*Figure 8C–D*). Future work will address the interplay between B1AR synthesis and its co-translational engagement with the translocon to ascertain whether there is a direct handoff between the translocon and the EMC or the EMC acts post-translationally to insert the N-terminal helix of B1AR.

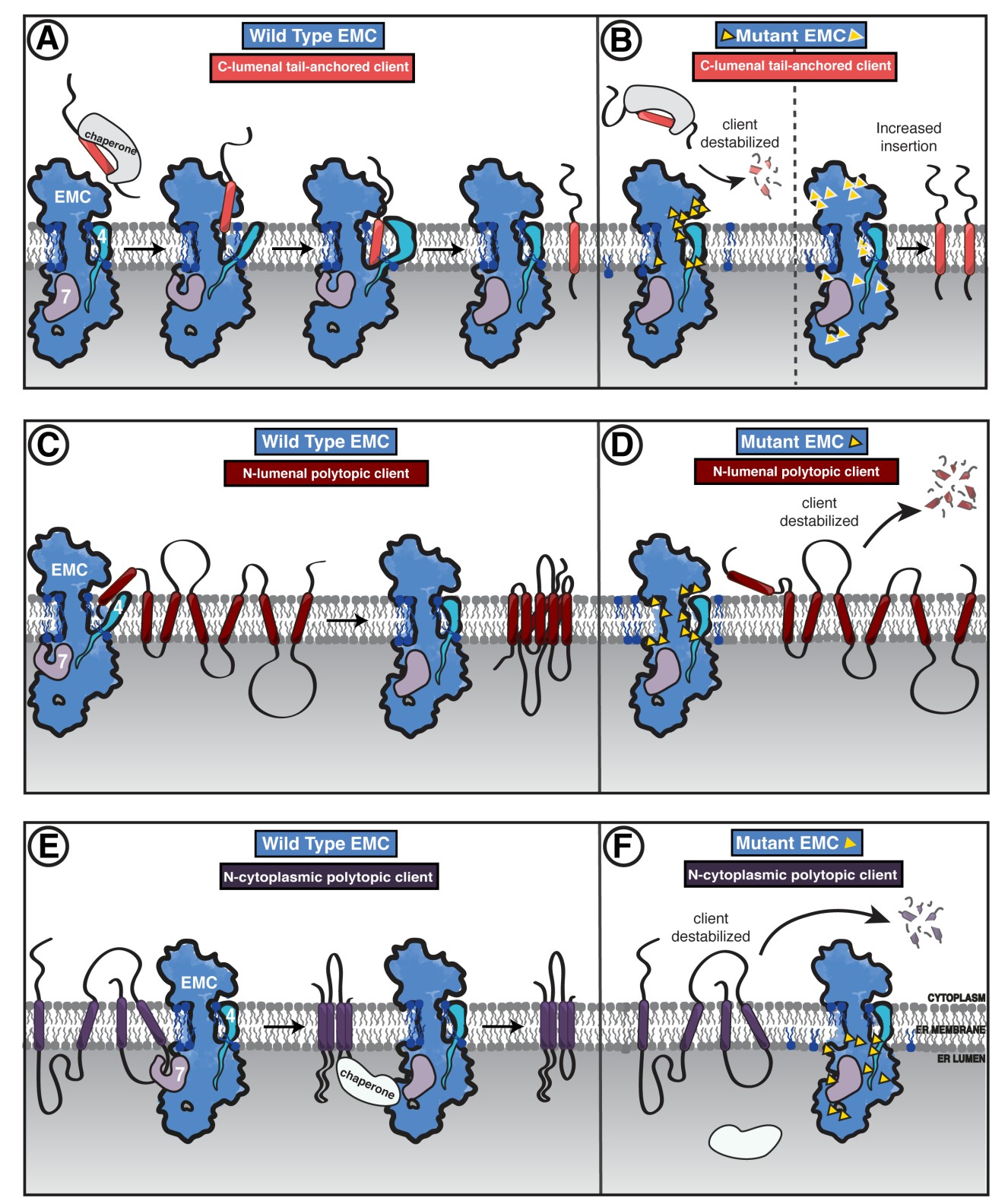

**Figure 8.** Model of coordinated EMC functions. (**A**) Model of EMC insertase function for a C-lumenal tail-anchored client. Cytosolic factors bring post-translationally localized clients to the ER. Then the client engages the EMC cytoplasmic domain. The polar roof modulates entry into the gated cavity. A hydrophobic slide facilitates the client helix fully entering the cavity. A lateral movement of the gate releases the client helix into the membrane and the EMC gate closes. (**B**) Our mutagenesis data provide the following insights into EMC regions of functional importance for each of the three client types

*Figure 8 continued on next page*

*Figure 8 continued*

we tested. Mutants are depicted by yellow triangles. Tail-anchored client (coral) abundance was depleted upon mutagenesis of the cytoplasmic domain entrance to the gated cavity, polar and charged residues at the cytoplasm-membrane boundary, residues along the length of the gated cavity, in the hydrophobic seal to the lumen, and lipid interacting residues in both cavities (left). We also observed a subset of mutants that resulted in higher levels of the C-lumenal tail-anchored client (right) that are positioned in the cytoplasmic domain cap, throughout the ER lumenal domain, and one mutation at the center of the gated cavity. (C) The EMC facilitates biogenesis of N-lumenal polytopic client protein B1AR (dark red). (D) Regions important for B1AR stability primarily map to the transmembrane region of the EMC structure, with depletion observed for lipid proximal residues on both sides of the cavity, the polar entrance roof of the gated cavity, and the EMC1 brace helix. (E) The EMC facilitates biogenesis of N-cytoplasmic polytopic client protein TMEM97 (dark purple). (F) Regions important for TMEM97 stability were primarily located in the lumenal domain spanning both propellers, in EMC1. In addition to these lumenal regions, there was a depletion of TMEM97 at the lipid-interacting positions at the lumenal interface of both membrane cavities of the EMC. Figure - Figure Supplement legends.

## The EMC lumenal domain orchestrates holdase chaperone function important for polytopic clients

Unlike the two terminal insertase clients we investigated, TMEM97 biogenesis was negatively impacted by mutation of the lumenal EMC1. The depletion of TMEM97 observed in these mutant backgrounds is consistent with the lumenal domain contributing to a holdase chaperone function, passively shielding its client while it is being synthesized and/or folded (*Zhang et al., 2017*). Interestingly, the diametrically opposed phenotype of mutants in the EMC lumenal domain on SQS raises the possibility that occupancy by one type of client can support an EMC conformation that is unfavorable for receiving the other. Alternative conformations could establish competition between client types for EMC occupancy. One explanation for this observation is that there is a conformational change between the insertase-active versus the holdase-active states. Interestingly, we identified at least two EMC conformations in our collection of structures, and EMC may adopt different conformations in various client and cofactor-engaged states.

In yeast, the polytopic clients co-purifying with the EMC are also glycosylated. One possible model is that the putative carbohydrate-binding domains in EMC7 or EMC10 directly contribute to engagement with client proteins. We speculate post-translational modifications on clients and the EMC could modulate function including client binding, chaperone binding, or regulating signaling in response to cellular cues.

Multi-pass transmembrane proteins require membrane factors to assist after insertion into the membrane to pack transmembrane helices in the correct order and topology. We propose that the EMC may act as a chaperone holdase to facilitate one of the following: helix and lipid packing, shielding from degradation while synthesis is in progress, or assisting in the assembly of multi-protein transmembrane complex formation. This is consistent with observations that in the absence of the EMC numerous integral membrane proteins are degraded (*Shurtleff et al., 2018*; *Volkmar et al., 2019*; *Tian et al., 2019*). Direct interactions with multi-pass transmembrane proteins have been shown previously (*Shurtleff et al., 2018*; *Coelho et al., 2019*). Furthermore, EMC dependence of internal transmembrane domain segments has also been established (*Ngo et al., 2019*; *Hiramatsu et al., 2019*). In the absence of yEMC7, a primarily lumenal subunit, a polytopic membrane protein was retained for longer in the ER, suggesting the possibility that yEMC7 may be involved in client release from the EMC. We propose a model where the EMC engages polytopic clients either during or directly after translation and remains bound until the client is released either to the membrane environment directly or handed off to client-specific and general ER chaperones (*Figure 8E–F*). It remains to be seen whether these polytopic clients directly engage with the lipid-filled cavity or the gated cavity or the lumen domain, what the determinants for engaging with a client or release into the membrane are, and how the EMC fits into the broader ER lumenal chaperone network.

## Potential role of the EMC as a master regulator of membrane protein biogenesis as the basis for its pleiotropic phenotypes

Why does the cell use a multifunctional EMC molecular machinery rather than specialized machinery for each of the functions encompassed by the EMC? Considering that the cell already has general machinery (Sec61 translocon) and tail-anchor insertase machinery (GET/TRC complex), we speculate that the EMC coordinates biogenesis of diverse membrane proteins. Several observations suggest

broader roles of the EMC as an integrator of information sensing the protein and lipid environment and coordinating its multiple activities, including the regulating the biogenesis of membrane proteins. For example, the initial identification of the EMC included numerous genetic interactions with both protein and lipid synthesis factors in yeast (*Jonikas et al., 2009*) and these disparate interdependencies have been subsequently observed in numerous species including human EMC (*Lahiri et al., 2014*; *Tang et al., 2017*; *Guna et al., 2018*; *Volkmar et al., 2019*; *Volkmar and Christianson, 2020*). Also, several client proteins are enzymes or cofactors involved in multiple stages of lipid synthesis or trafficking, and this may provide a unifying explanation for the range of genetic interactions and co-essentiality observations reported to date (*Guna et al., 2018*; *Shurtleff et al., 2018*; *Volkmar et al., 2019*; *Tian et al., 2019*; *Wainberg et al., 2019*; *Corradi et al., 2019*; *Volkmar and Christianson, 2020*). Perhaps by facilitating the insertion of sterol synthesis protein SQS, the EMC allows for modulation of local membrane thickness and lipid composition to accommodate differences within the broad range of membrane proteins being synthesized. In this regard, one structural feature of particular interest is the EMC1 amphipathic brace, which resides adjacent to the lipid-filled cavity. This conserved feature sits within the interfacial membrane boundary, raising the possibility that it can modulate the lipid or protein composition of this cavity. Notably, several other membrane proteins involved in ER homeostasis, including Opi1 and Ire1, also contain amphipathic helices that have been proposed to sense the properties of the lipid bilayer (*Volmer et al., 2013*; *Jacquemyn et al., 2017*; *Halbleib et al., 2017*; *Hofbauer et al., 2018*; *Cho et al., 2019*). Future work will explore how the EMC overall, and the EMC1 brace helix in particular, govern client release into the membrane, interface with the local structure of the lipid bilayer, and play roles in specific client-lipid interactions.

In addition to the three client classes we investigate here, it is clear that EMC has a broader range of clients including multi-protein assemblies (*Richard et al., 2013*; *Talbot et al., 2019*), lipid-modulating proteins (*Volkmar et al., 2019*), lipid-binding proteins (*Salas-Estrada et al., 2018*; *Sejdiu and Tieleman, 2020*), and those with helices that do not span the bilayer (*Lin et al., 2019*; *Ngo et al., 2019*). The compartmentalization and interdependence that we observe for effects of mutations on client handling provide a foundation for understanding this multifunctionality. We propose that the complexity of the EMC machine, combining insertase and holdase chaperone functions within one molecular machine, has arisen to mitigate the error prone biogenesis of a diverse range of membrane spanning proteins in the dynamic environment of the ER.

# Materials and methods

**Key resources table**

| Reagent type (species) or resource | Designation | Source or reference | Identifiers | Additional information |
|---|---|---|---|---|
| Gene (*Homo sapiens*) | hEMC1 | NIH Mammalian Gene Collection | NCBI: BC034589 | |
| Gene (*Homo sapiens*) | hEMC2 | NIH Mammalian Gene Collection | NCBI: BC021667 | |
| Gene (*Homo sapiens*) | hEMC3 | NIH Mammalian Gene Collection | NCBI: BC022807 | |
| Gene (*Homo sapiens*) | hEMC4 | Genestrand (Eurofins, Germany) | Uniprot: Q5J8M3-1 | |
| Gene (*Homo sapiens*) | hEMC5 | NIH Mammalian Gene Collection | NCBI: BC033588 | |
| Gene (*Homo sapiens*) | hEMC6 | NIH Mammalian Gene Collection | NCBI: BC001409 | |
| Gene (*Homo sapiens*) | hEMC7 | NIH Mammalian Gene Collection | NCBI: BC104936 | |
| Gene (*Homo sapiens*) | hEMC8 | NIH Mammalian Gene Collection | NCBI: BC020250 | |
| Gene (*Homo sapiens*) | hEMC9 | NIH Mammalian Gene Collection | NCBI: BC002491 | |

*Continued on next page*

*Continued*

| Reagent type (species) or resource | Designation | Source or reference | Identifiers | Additional information |
|---|---|---|---|---|
| Gene (*Homo sapiens*) | hEMC10 | Genestrand (Eurofins, Germany) | Uniprot: Q5UCC4-1 | |
| Gene (*Saccharomyces cerevisiae*) | yEMC1 | Uniprot | Uniprot: P25574 | |
| Gene (*Saccharomyces cerevisiae*) | yEMC2 | Uniprot | Uniprot: P47133 | |
| Gene (*Saccharomyces cerevisiae*) | yEMC3 | Uniprot | Uniprot: P36039 | |
| Gene (*Saccharomyces cerevisiae*) | yEMC4 | Uniprot | Uniprot: P53073 | |
| Gene (*Saccharomyces cerevisiae*) | yEMC5 | Uniprot | Uniprot: P40540 | |
| Gene (*Saccharomyces cerevisiae*) | yEMC6 | Uniprot | Uniprot: Q12431 | |
| Gene (*Saccharomyces cerevisiae*) | yEMC7 | Uniprot | Uniprot: P39543 | |
| Gene (*Saccharomyces cerevisiae*) | yEMC10 | Uniprot | Uniprot: Q12025 | |
| Recombinant DNA reagent | pX458 | Addgene | pX458 | |
| Recombinant DNA reagent | pKDP041 | This study; available from the Weissman Lab | Cas9-sfGFP-EMC5 sgRNA3 | single guide KO system targeting EMC5 gene |
| Recombinant DNA reagent | pKDP077 | This study; available from the Weissman Lab | Cas9-sfGFP-EMC1_sgRNA3_sgRNA4 | dual guide KO system targeting EMC1 gene |
| Recombinant DNA reagent | pKDP080 | This study; available from the Weissman Lab | Cas9-sfGFP-EMC2_sgRNA4_sgRNA5 | dual guide KO system targeting EMC2 gene |
| Recombinant DNA reagent | pKDP083 | This study; available from the Weissman Lab | Cas9-sfGFP-EMC3_sgRNA1_sgRNA2 | dual guide KO system targeting EMC3 gene |
| Recombinant DNA reagent | pKDP119 | This study; available from the Weissman Lab | SFFV-insert site-IRES-Puro-P2A-BFP | parental vector |
| Recombinant DNA reagent | pKDP121 | This study; available from the Weissman Lab | pTwist+Lenti+SFFV+EMC1+IRES+Puro+P2A+BFP+WPRE | EMC1 covering plasmid |
| Recombinant DNA reagent | pKDP122 | This study; available from the Weissman Lab | pTwist+Lenti+SFFV+EMC3+IRES+Puro+P2A+BFP+WPRE | EMC3 covering plasmid |
| Recombinant DNA reagent | pKDP124 | This study; available from the Weissman Lab | pTwist+Lenti+SFFV+EMC5+IRES+Puro+P2A+BFP+WPRE | EMC5 covering plasmid |
| Recombinant DNA reagent | pKDP125 | This study; available from the Weissman Lab | pTwist+Lenti+SFFV+EMC2+IRES+Puro+P2A+BFP+WPRE | EMC2 covering plasmid |
| Recombinant DNA reagent | pKDP110 | This study; available from the Weissman Lab | bAR1_mCherry_P2A_GFP | See *Supplementary file 5* for sequence |
| Recombinant DNA reagent | pKDP111 | This study; available from the Weissman Lab | TMEM97_mCherry_P2A_GFP | See *Supplementary file 5* for sequence |
| Recombinant DNA reagent | pKDP136 | This study; available from the Weissman Lab | GFP_P2A_mCherry_SQS_TMD_opsintag | See *Supplementary file 5* for sequence |
| Recombinant DNA reagent | pKDP119_hsEMC1_mut_D31K | Twist; available from the Weissman Lab | hsEMC1_mut_D31K | See *Supplementary file 5* for sequence |
| Recombinant DNA reagent | pKDP119_hsEMC1_mut_R69D | Twist; available from the Weissman Lab | hsEMC1_mut_R69D | See *Supplementary file 5* for sequence |
| Recombinant DNA reagent | pKDP119_hsEMC1_mut_G71S | Twist; available from the Weissman Lab | hsEMC1_mut_G71S | See *Supplementary file 5* for sequence |

*Continued on next page*

*Continued*

| Reagent type (species) or resource | Designation | Source or reference | Identifiers | Additional information |
|---|---|---|---|---|
| Recombinant DNA reagent | pKDP119_hsEMC1_mut_R76D_K80D | Twist; available from the Weissman Lab | hsEMC1_mut_R76D_K80D | See *Supplementary file 5* for sequence |
| Recombinant DNA reagent | pKDP119_hsEMC1_mut_T82M | Twist; available from the Weissman Lab | hsEMC1_mut_T82M | See *Supplementary file 5* for sequence |
| Recombinant DNA reagent | pKDP119_hsEMC1_mut_T82A | Twist; available from the Weissman Lab | hsEMC1_mut_T82A | See *Supplementary file 5* for sequence |
| Recombinant DNA reagent | pKDP119_hsEMC1_mut_A144T | Twist; available from the Weissman Lab | hsEMC1_mut_A144T | See *Supplementary file 5* for sequence |
| Recombinant DNA reagent | pKDP119_hsEMC1_mut_H93D_E138D_N282K | Twist; available from the Weissman Lab | hsEMC1_mut_H93D_E138D_N282K | See *Supplementary file 5* for sequence |
| Recombinant DNA reagent | pKDP119_hsEMC1_mut_R275E_R404E | Twist; available from the Weissman Lab | hsEMC1_mut_R275E_R404E | See *Supplementary file 5* for sequence |
| Recombinant DNA reagent | pKDP119_hsEMC1_mut_G471R | Twist; available from the Weissman Lab | hsEMC1_mut_G471R | See *Supplementary file 5* for sequence |
| Recombinant DNA reagent | pKDP119_hsEMC1_mut_F473Y_R487K | Twist; available from the Weissman Lab | hsEMC1_mut_F473Y_R487K | See *Supplementary file 5* for sequence |
| Recombinant DNA reagent | pKDP119_hsEMC1_mut_M483A_R487H_Q491N | Twist; available from the Weissman Lab | hsEMC1_mut_M483A_R487H_Q491N | See *Supplementary file 5* for sequence |
| Recombinant DNA reagent | pKDP119_hsEMC1_mut_G868R | Twist; available from the Weissman Lab | hsEMC1_mut_G868R | See *Supplementary file 5* for sequence |
| Recombinant DNA reagent | pKDP119_hsEMC1_mut_R881C | Twist; available from the Weissman Lab | hsEMC1_mut_R881C | See *Supplementary file 5* for sequence |
| Recombinant DNA reagent | pKDP119_hsEMC1_mut_K951A_K957A | Twist; available from the Weissman Lab | hsEMC1_mut_K951A_K957A | See *Supplementary file 5* for sequence |
| Recombinant DNA reagent | pKDP119_hsEMC2_mut_K18A_K21A | Twist; available from the Weissman Lab | hsEMC2_mut_K18A_K21A | See *Supplementary file 5* for sequence |
| Recombinant DNA reagent | pKDP119_hsEMC2_mut_R80A_R81A_K90A_R112A | Twist; available from the Weissman Lab | hsEMC2_mut_R80A_R81A_K90A_R112A | See *Supplementary file 5* for sequence |
| Recombinant DNA reagent | pKDP119_hsEMC2_mut_K125E_R126D_K127E | Twist; available from the Weissman Lab | hsEMC2_mut_K125E_R126D_K127E | See *Supplementary file 5* for sequence |
| Recombinant DNA reagent | pKDP119_hsEMC2_mut_N137A_N167A | Twist; available from the Weissman Lab | hsEMC2_mut_N137A_N167A | See *Supplementary file 5* for sequence |
| Recombinant DNA reagent | pKDP119_hsEMC2_mut_E146A_E149A_Q150A | Twist; available from the Weissman Lab | hsEMC2_mut_E146A_E149A_Q150A | See *Supplementary file 5* for sequence |
| Recombinant DNA reagent | pKDP119_hsEMC2_mut_E168A_D170A_K173A | Twist; available from the Weissman Lab | hsEMC2_mut_E168A_D170A_K173A | See *Supplementary file 5* for sequence |
| Recombinant DNA reagent | pKDP119_hsEMC2_mut_E206A_E209A_D252A | Twist; available from the Weissman Lab | hsEMC2_mut_E206A_E209A_D252A | See *Supplementary file 5* for sequence |
| Recombinant DNA reagent | pKDP119_hsEMC2_mut_K248E_D252K_K255E | Twist; available from the Weissman Lab | hsEMC2_mut_K248E_D252K_K255E | See *Supplementary file 5* for sequence |
| Recombinant DNA reagent | pKDP119_hsEMC2_mut_R266A_Q269A_R273A | Twist; available from the Weissman Lab | hsEMC2_mut_R266A_Q269A_R273A | See *Supplementary file 5* for sequence |
| Recombinant DNA reagent | pKDP119_hsEMC2_mut_Q269A_E286A_E290A | Twist; available from the Weissman Lab | hsEMC2_mut_Q269A_E286A_E290A | See *Supplementary file 5* for sequence |
| Recombinant DNA reagent | pKDP119_hsEMC3_WT | Twist; available from the Weissman Lab | hsEMC3_WT | See *Supplementary file 5* for sequence |
| Recombinant DNA reagent | pKDP119_hsEMC3_mut_D9A | Twist; available from the Weissman Lab | hsEMC3_mut_D9A | See *Supplementary file 5* for sequence |
| Recombinant DNA reagent | pKDP119_hsEMC3_mut_R13E | Twist; available from the Weissman Lab | hsEMC3_mut_R13E | See *Supplementary file 5* for sequence |

*Continued on next page*

*Continued*

| Reagent type (species) or resource | Designation | Source or reference | Identifiers | Additional information |
|---|---|---|---|---|
| Recombinant DNA reagent | pKDP119_hsEMC3_mut_K42A_K43A | Twist; available from the Weissman Lab | hsEMC3_mut_K42A_K43A | See *Supplementary file 5* for sequence |
| Recombinant DNA reagent | pKDP119_hsEMC3_mut_E63K_D213K_E223K | Twist; available from the Weissman Lab | hsEMC3_mut_E63K_D213K_E223K | See *Supplementary file 5* for sequence |
| Recombinant DNA reagent | pKDP119_hsEMC3_mut_K70Y | Twist; available from the Weissman Lab | hsEMC3_mut_K70Y | See *Supplementary file 5* for sequence |
| Recombinant DNA reagent | pKDP119_hsEMC3_mut_V118A_I122A | Twist; available from the Weissman Lab | hsEMC3_mut_V118A_I122A | See *Supplementary file 5* for sequence |
| Recombinant DNA reagent | pKDP119_hsEMC3_mut_N114D_N117D | Twist; available from the Weissman Lab | hsEMC3_mut_N114D_N117D | See *Supplementary file 5* for sequence |
| Recombinant DNA reagent | pKDP119_hsEMC3_mut_R180A | Twist; available from the Weissman Lab | hsEMC3_mut_R180A | See *Supplementary file 5* for sequence |
| Recombinant DNA reagent | pKDP119_hsEMC3_mut_R59E_R62E_K216E | Twist; available from the Weissman Lab | hsEMC3_mut_R59E_R62E_K216E | See *Supplementary file 5* for sequence |
| Recombinant DNA reagent | pKDP119_hsEMC3_mut_R147E | Twist; available from the Weissman Lab | hsEMC3_mut_R147E | See *Supplementary file 5* for sequence |
| Recombinant DNA reagent | pKDP119_hsEMC3_mut_F148L | Twist; available from the Weissman Lab | hsEMC3_mut_F148L | See *Supplementary file 5* for sequence |
| Recombinant DNA reagent | pKDP119_hsEMC3_mut_M151L | Twist; available from the Weissman Lab | hsEMC3_mut_M151L | See *Supplementary file 5* for sequence |
| Recombinant DNA reagent | pKDP119_hsEMC3_mut_I186V_I182V | Twist; available from the Weissman Lab | hsEMC3_mut_I186V_I182V | See *Supplementary file 5* for sequence |
| Recombinant DNA reagent | pKDP119_hsEMC3_mut_K244A_H247A_E249A | Twist; available from the Weissman Lab | hsEMC3_mut_K244A_H247A_E249A | See *Supplementary file 5* for sequence |
| Recombinant DNA reagent | pKDP119_hsEMC5_WT | Twist; available from the Weissman Lab | hsEMC5_WT | See *Supplementary file 5* for sequence |
| Recombinant DNA reagent | pKDP119_hsEMC5_mut_A18L | Twist; available from the Weissman Lab | hsEMC5_mut_A18L | See *Supplementary file 5* for sequence |
| Recombinant DNA reagent | pKDP119_hsEMC5_mut_D44K | Twist; available from the Weissman Lab | hsEMC5_mut_D44K | See *Supplementary file 5* for sequence |
| Recombinant DNA reagent | pKDP119_hsEMC5_mut_D82A_R85A | Twist; available from the Weissman Lab | hsEMC5_mut_D82A_R85A | See *Supplementary file 5* for sequence |
| Recombinant DNA reagent | pKDP119_hsEMC5_mut_F22L | Twist; available from the Weissman Lab | hsEMC5_mut_F22L | See *Supplementary file 5* for sequence |
| Recombinant DNA reagent | pKDP119_hsEMC5_mut_E75A | Twist; available from the Weissman Lab | hsEMC5_mut_E75A | See *Supplementary file 5* for sequence |
| Recombinant DNA reagent | pKDP119_hsEMC5_mut_H19L_S23A_Q26L | Twist; available from the Weissman Lab | hsEMC5_mut_H19L_S23A_Q26L | See *Supplementary file 5* for sequence |
| Recombinant DNA reagent | pKDP119_hsEMC5_mut_K7A | Twist; available from the Weissman Lab | hsEMC5_mut_K7A | See *Supplementary file 5* for sequence |
| Recombinant DNA reagent | pKDP119_hsEMC5_mut_K7E | Twist; available from the Weissman Lab | hsEMC5_mut_K7E | See *Supplementary file 5* for sequence |
| Recombinant DNA reagent | pKDP119_hsEMC5_mut_R28A_R32A | Twist; available from the Weissman Lab | hsEMC5_mut_R28A_R32A | See *Supplementary file 5* for sequence |
| Recombinant DNA reagent | pKDP119_hsEMC5_mut_I63L | Twist; available from the Weissman Lab | hsEMC5_mut_I63L | See *Supplementary file 5* for sequence |
| Recombinant DNA reagent | pKDP119_hsEMC5_mut_F90A | Twist; available from the Weissman Lab | hsEMC5_mut_F90A | See *Supplementary file 5* for sequence |
| Antibody | Mouse GAPDH Primary Antibody | Abcam | ab8245 | See *Supplementary file 5* for sequence |
| Antibody | Rabbit TMEM97 primary | ThermoFisher Scientific | PA-23003 | |

*Continued on next page*

*Continued*

| Reagent type (species) or resource | Designation | Source or reference | Identifiers | Additional information |
|---|---|---|---|---|
| Antibody | Rabbit FDFT1 Primary Antibody | Abcam | ab195046 | |
| Antibody | Rat BAP31 Primary Antibody | ThermoFisher Scientific | MA3-002 | |
| Antibody | Rabbit (KIAA0090) EMC1 primary antibody | Abcam | ab242112 | |
| Antibody | Rabbit TTC35 (EMC2) primary antibody | Proteintech | 25443–1-AP | |
| Antibody | Rabbit TM111 (EMC3) primary antibody | ThermoFisher Scientific | #711771 | |
| Antibody | Rabbit EMC4 primary antibody | Abcam | ab184544 | |
| Antibody | Rabbit MMGT1 (EMC5) primary antibody | Bethyl Laboratories | A305-833A-M | |
| Antibody | Rabbit (C19orf63) EMC10 primary antibody | Abcam | ab180148 | |
| Antibody | IRDye 800CW Goat anti-Mouse IgG Secondary Antibody | LI-COR Biosciences | 925–32210 | |
| Antibody | IRDye 800CW Goat anti-Rabbit IgG Secondary Antibody | LI-COR Biosciences | 926–32211 | |
| Peptide, recombinant protein | Fab DE4 | This study; available from the Weissman Lab | LMV83 | LFAIPLVVPFYSHSALDVVMTQSPLSLPV TPGEPASISCRSSQTLMNRNGNNFLDW YVQKPGQSPQLLIYLGSNRAPGVPDRFS GSGSGTDFTLKISRLEVEDVGVYYCMQA LQTPRTFGQGTKVEIKRTVAAPSVFIFPP SDEQLKSGTASVVCLLNNFYPREAKVQW KVDNALQSGNSQESVTEQDSKDSTYSLS STLTLSKADYEKHKVYACEVTHQGLSSP VTKSFNRGEC– MAQVQLQQWGAGLLKPSETLSLTCAVYG GSFSGYYWSWIRQPPGKGLEWIGEINHS GSTNYNPSLKSRVTISVDTSKKQFSLKLS SVTAADTAVYYCARFSYYGSGIYWGQGTL VTVSSASTKGPSVFPLAPSSKSTSGGTAA LGCLVKDYFPEPVTVSWNSGALTSGVHT FPAVLQSSGLYSLSSVVTVPSSSLGTQTYI CNVNHKPSNTKVDKKVEPKSCAAAHHH HHHGAAEQKLISEEDLNGAA- |
| Peptide, recombinant protein | Fab DH4 | This study; available from the Weissman Lab | LMV82 | LFAIPLVVPFYSHSALDVVMTQSPLSLPV TPGEPASISCRSSQTLMNRNGNNFLDW YLQKPGQSPQLLIYLGSNRAPGVPDRFS GSGSGTDFTLRISRVEPEDVGVYYCMQA LQTPSFGGGTKVEIRRTVAAPSVFIFPPS DEQLKSGTASVVCLLNNFYPREAKVQW KVDNALQSGNSQESVTEQDSKDSTYSL SSTLTLSKADYEKHKVYACEVTHQGLSS PVTKSFNRGEC– MAQVQLQQWGAGLLKPSETLSLTCAVY GGSFSGYYWSWIRQPPGKGLEWIGEIN HSGSTNYNPSLKSRVTISVDTSKNQFSL KLSSVTAADTAVYYCARGLAGRGYYGSG SYLRWGQGTLVTVSSASTKGPSVFPLAP SSKSTSGGTAALGCLVKDYFPEPVTVSW NSGALTSGVHTFPAVLQSSGLYSLSSVV TVPSSSLGTQTYICNVNHKPSNTKVDK KVEPKSCAAAHHHHHHGAAE QKLISEEDLNGAA- |
| Commercial assay or kit | Superose 6, 10/300 GL | GE Healthcare | 17517201 | |

*Continued on next page*

*Continued*

| Reagent type (species) or resource | Designation | Source or reference | Identifiers | Additional information |
|---|---|---|---|---|
| Commercial assay or kit | R1.2/1.3 200 and 300 mesh Cu holey carbon grids | Quantifoil | 1210627 | |
| Commercial assay or kit | BL21 Gold Star competent cells | Invitrogen | C602003 | |
| Commercial assay or kit | Anti-Flag agarose beads | Millipore | A2220 | |
| Commercial assay or kit | EconoPac Chromatography Columns | Biorad | 7321010 | |
| Commercial assay or kit | 100 KD MW | EMD Millipore | UFC810024 | |
| Commercial assay or kit | Superose 6, 10/300 GL | Cytiva | 29-0915-96 | |
| Commercial assay or kit | cOmplete EDTA-free Protease Inhibitor Cocktail | Roche | catalog No. 05056489001 | |
| Commercial assay or kit | Bio-Beads | Biorad | 1523920 | |
| Commercial assay or kit | R1.2/1.3 200 and 300 mesh Cu holey carbon grids | Quantifoil | 1210627 | |
| Commercial assay or kit | Ultrathin Carbon Film on Lacey Carbon Support Film, 400 mesh, Copper | Ted Pella | #01824 | |
| Chemical compound, drug | FuGENE HD transfection reagent | Promega | E2312 | |
| Chemical compound, drug | 1-Palmitoyl-2-oleoyl-sn-glycero-3-PC (POPC) | Cayman Chemical | 15102 | |
| Chemical compound, drug | Glyco-diosgenin (GDN) | Anatrace | GDN101 | |
| Chemical compound, drug | yeast extract total | Avanti Polar Lipids | 190000 P-100mg | |
| Chemical compound, drug | Cholesteryl Hemisuccinate Tris Salt | Anatrace | CH210 5 GM | |
| Chemical compound, drug | b-DDM | Anatrace | D310 | |
| Chemical compound, drug | IPTG | GoldBio | I2481C5 | |
| Chemical compound, drug | EX-CELL 420 Serum-Free Medium | Sigma-Aldrich | 14420 C | |
| Chemical compound, drug | FreeStyle 293 Expression Medium | Thermo fischer | 12338018 | |
| Cell line (*Homo sapiens*) | HEK293S GnTI- | ATCC | CRL-3022 | Mycoplasma negative |
| Cell line (*Spodoptera frugiperda*) | Sf9 | Thermo Fischer | 11496015 | |
| Cell line (*Homo sapiens*) | K562 crispri | *Gilbert et al., 2014* | K562 crispri | |

*Continued on next page*

*Continued*

| Reagent type (species) or resource | Designation | Source or reference | Identifiers | Additional information |
|---|---|---|---|---|
| Strain, strain background *Saccharomyces cerevisiae* | Overexpressed EMC with yEMC5-linker-TEV-linker-3xFlag | This study; available from the Weissman Lab | LMV84 | BY4743 —— MATa/alpha, his3Δ0/his3Δ0, leu2Δ0/leu2Δ0, LYS2/lys2Δ0, met15Δ0/MET15, ura3Δ0/ura3Δ0, emc1::NatMX::TEF2pr-EMC1/EMC1, emc3::KanMX::TEF2pr-EMC3/EMC3, emc4::his3(CG)::TEF2pr-EMC4/EMC4, sop4::HphMx::TEF2pr-SOP4/SOP4, EMC2/emc2::NatMX::TEF2pr-EMC2, emc5::EMC5-TEV-3xFLAG::ura3(KL)/emc5::his3(CG)::TEF2pr-EMC5-TEV-3xFLAG::KanMX, EMC6/emc6::HphMX::TEF2pr-EMC6, YDR056c/ydr056c::leu2(CG)::TEF2pr-ydr056c |
| Strain, strain background *Saccharomyces cerevisiae* | Endogenous yEMC5-linker-TEV-linker-3xFlag | This study; available from the Weissman Lab | LMV85 | W303 —— EMC5-3xF:ura - Linker-TEV-linker-3xFlag (GGSGSGENLYFQSGSGS DYKDDDDKDYKDDDDKDYKDDDDK) |
| Software, algorithm | CryoSPARC version 2.12.4. | *Punjani et al., 2017* | RRID:SCR_016501 | |
| Software, algorithm | UCSF ChimeraX Version 1.0 | *Goddard et al., 2018* | RRID:SCR_015872 | |
| Software, algorithm | PHENIX Version 1.17 | *Adams et al., 2011*; | RRID:SCR_014224 | |
| Software, algorithm | *Coot Version 0.8* | *Emsley et al., 2010* | RRID:SCR_014222 | |
| Software, algorithm | RELION 3.1 | *Kimanius et al., 2016*; *Zivanov et al., 2018* | http://www2.mrclmb.cam.ac.uk/relion | |
| Software, algorithm | SerialEM | *Mastronarde, 2005* | RRID:SCR_017293 | |

Reagents used for experiments described and reagents made as part of this study are listed in a Key Resources Table listed as an appendix to this article file.

## Cell line maintenance

K562 dCas9 KRAB cells were grown in RPMI 1640 (GIBCO) with 25 mM HEPES, 2 mM l-glutamine, 2 g/L NaHCO$_3$ and supplemented with 10% (v/v) fetal bovine serum (FBS), 100 units/mL penicillin, 100 μg/mL streptomycin, 2 mM l-glutamine. HEK293T cells were grown in Dulbecco's modified eagle medium (DMEM, GIBCO) with 25 mM d-glucose, 3.7 g/L NaHCO$_3$, 4 mM l-glutamine and supplemented with 10% (v/v) FBS, 100 units/mL penicillin, 100 μg/mL streptomycin. All cell lines were grown at 37°C. All cell lines were periodically tested for Mycoplasma contamination using the MycoAlert Plus Mycoplasma detection kit (Lonza).

## DNA transfections and virus production

Lentivirus was generated by transfecting HEK39T cells with standard fourth-generation packaging vectors using TransIT-LT1 Transfection Reagent (Mirus Bio). Media was changed 10 hr post-transfection. Viral supernatant was harvested 60 hr after transfection, filtered through 0.45 μm PVDF filters and frozen prior to transduction.

## Knockout hEMC cell lines

A single and dual knockout guide system was developed in the pX458 backbone (Addgene plasmid # 48138) with guides targeting hEMC1, hEMC2, hEMC3, or hEMC5 (Key Resources table). Targeting guides were selected using the Broad's guide selection tool (https://portals.broadinstitute.org/gpp/public/analysis-tools/sgrna-design). For the single hEMC5 knockout system, an hEMC5 targeting guide was cloned into pX458 by digesting with BbsI and ligating to annealed oligos for the hEMC5 sgRNA. For the dual knockout system, a four-step cloning process generated the final knockout

plasmid: (1) Each of the two guides targeting the same locus were individually cloned into pX458. (2) Then pX458_sgRNA1 was digested with XbaI. (3) SgRNA2 cassette from pX458_sgRNA2 was PCR amplified with oligos containing overhangs spanning the XbaI cloning site and purified. (4) Finally, the final dual guide vector was generated by Gibson cloning (NEBuilder).

To generate the hEMC knockout cell lines, K562 dCas9 KRAB cells were nucleofected with the respective hEMC knockout plasmids using Lonza SF Cell Line 96-well Nucleofector Kit (V4SC-2096). Two days post-nucleofection, GFP-positive cells were single cell sorted into 96-well plates using BD FACS AriaII. After colonies from single cells grew out, genomic DNA was isolated using QuickExtract (Lucigen), the sgRNA-targeted sites were PCR amplified and then NGS-sequenced via Genewiz's EZ-Amplicon service. Sequencing data was analyzed and aligned to the respective reference alleles in the human genome. Clones whose alleles harbored only indel mutations for hEMC1, hEMC2, hEMC3, and hEMC5 (full knockouts) respectively were further validated on the protein level.

### Dual fluorescent EMC client reporter cell lines

Dual client reporters for TMEM97, ADRB1 (protein name: B1AR), and FDFT1 (protein name: SQS) were introduced lentivirally into each of the EMC1, EMC2, EMC3, and EMC5 knockout cell lines. TMEM97 and ADRB1 full-length sequences were used with a C-terminal tag -mCherry-P2A-GFP. The sequence for FDFT1 transmembrane domain (SQS$^{378-410}$) was tagged N-terminally with GFP-P2A-mCherry- and an opsin tag on the C-terminus as used in a prior study (*Guna et al., 2018*). Three days post-transduction, GFP/mCherry-positive cells were sorted on BDAriaII. Sequences for these constructs are available in the *Supplementary file 5*.

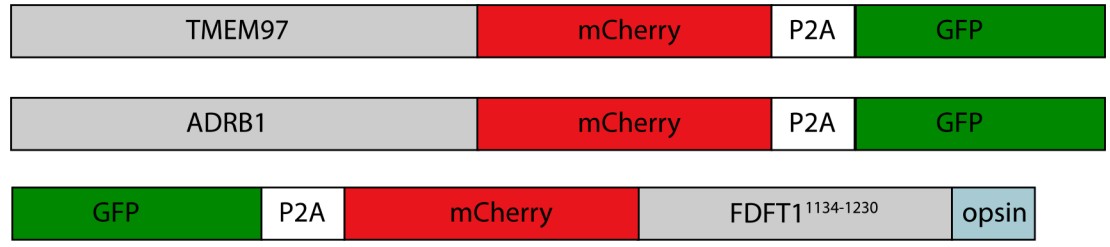

**Scheme 1.** Client reporters.

### Mutant EMC cell lines

The EMC mutant genes were synthesized and cloned by Twist into pKDP119-SFFV-[insert site]-IRES-Puro-P2A-BFP. For hEMC subunit mutation details refer to the Key Resources Table, for sequences refer to *Supplementary file 5*. Mutant hEMC cell lines were generated by lentiviral introduction of the respective hEMC mutant subunit into the respective knockout cell lines (hEMC1, hEMC2, hEMC3, or hEMC5) containing the dual fluorescent reporters for each EMC client (pKDP110_ADRB1_mCherry_P2A_GFP, pKDP111_TMEM97_mCherry_P2A_GFP, or GFP_P2A_m-Cherry_FDFT1_TMD_opsintag). The expression of each fluorescent reporter was read out 6 days after puromycin selection in each of the hEMC mutant cell lines.

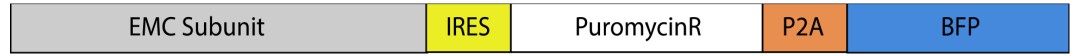

**Scheme 2.** hEMC subunit mutation construct design.

### Flow analysis

For each hEMC mutant cell line, 20,000 live cells were recorded on Attune NxT flow cytometer. FlowCal flow analysis package was used for analysis in Python. First, live cells were gated based on FSC/SSC. Then GFP (BL1-A) and mChery (YL2-A) were plotted for each mutant and control cell line. mCherry:GFP intensity ratios were calculated for individual cells in each cell line. Fluorescence ratios for each substrate in an hEMC mutant cell line were normalized to the mCherry:GFP ratio of the same substrate in the hEMC wild-type rescue cell line. Distributions of fluorescence ratios were plotted as histograms in Python using seaborn.

## Fluorescent reporter statistical analysis

We performed bootstrap estimates of the mean of normalized mCherry/GFP ratio from the FACS data. For bootstrapping, we performed 1000 iterations with 50 cells/iteration to fit normal distributions. We performed two separate one-sided T-tests at a p-value cutoff of 0.01 between each mutant and the respective subunit WT to test for significant decreases or increases in ratios based on bootstrapped estimates of the mean. These statistics are contained in the files 'filtered_final_pvalues.01cutoff.lo.csv' and 'filtered_final_pvalues.01cutoff.hi.csv' respectively. Statistics were generated for the EMC-independent membrane protein controls ('stats.membrane.controls.lo', 'stats.membrane.controls.hi') and for the mCherry-p2a-GFP control ('.mcherry.p2a.gfp.control.lo', 'stats.mcherry.p2a.gfp.control.hi'). Values can be found in *Supplementary file 2*.

## Western blotting

Cell pellets were lysed using lysis buffer (20 mM Tris pH 7.5, 150 mM NaCl, 5 $MgCl_2$, 1% Triton x-100, 1 mM DTT, 24 U/ml) Turbo DNase (Ambion). Clarified lysate was quantified and samples were boiled with 4x LDS sample (Thermo Fisher, NP0007) buffer for 5 min at 95℃. Samples were separated on 4–12% or 12% Bolt Bis-Tris Plus Gels (Invitrogen, NP0322PK2). Proteins were transferred onto nitrocellulose membranes using Bio-Rad Trans-Blot Turbo transfer system. Membranes were blocked in Odyssey Blocking Buffer (LI-COR, 927–50000) for an hour at room temperature. Blocked membranes were incubated with primary antibody diluted in TBST and incubated overnight at 4℃ on a shaker. Primary antibodies were detected by incubating membranes with 1:10,000 dilution of IRDye-conjugated (LI-COR) secondary anti-mouse and anti-rabbit antibodies for 1 hr at room temperature. Blots were visualized using LI-COR imaging system. The primary antibodies used in this study are listed in the Key Resources table.

## Yeast strains

Strain BY4741 and BY4742 were used as the wild-type parental strains for the creation of the yEMC overexpression strain. Yeast homologous recombination (*Rothstein, 1991*) was used to generate yeast strains. For the overexpression strain, the endogenous promotor for each yEMC subunit (yEMC1, yEMC2, yEMC3, yEMC4, yEMC5, yEMC6, yEMC7, yEMC10) were replaced with a TEF2 promoter. In addition, EMC5 was tagged at the C-terminus with linker-TEV-linker-3xFlag. Auxotrophic markers and drug selection markers in both BY4741 and BY4742 were employed to add this promoter modification to all these eight subunits and the two strains were crossed to create the resulting BY4743 strain used for immunoprecipitation. Endogenous EMC yeast strain was made using W303a wild-type parental background (leu2-3,−112; his3-11,−15; trp1-1; ura3-1; ade2-1; can1-100; MATa). Homologous recombination was used to integrate a linker-TEV-linker-3xFlag at the C-terminus of yEMC coding sequence. Genomic PCR was conducted to verify integration.

## Design and purification of fragments antigen binding (Fab) DH4 and DE4

Fabs were identified as described in these studies (*Kim et al., 2011*; *Wu et al., 2012*). Overexpressed yEMC solubilized in DDM as described above was biotinylated and streptavidin magnetic beads were used to capture yEMC, which was then subjected to a Fab phage library. Unbound Fabs were washed away and then binding Fabs were eluted and analyzed by ELISA. Two Fabs were identified binding yEMC, Fab DH4 and DE4.

## Purification of DH4 and DE4 Fabs

Plasmid with either Fab DH4 or DE4 were transformed into BL21 Gold Star cells and plated onto agarose plates with 2x YT + 2% glucose + Ampicillin. Cultures were inoculated from resulting colonies for overnight growth at 30℃ into 2xYT + 2% glucose + Amp. In the morning dilute overnight culture to $OD_{600}$ of 0.05 in 1L, in a 2.8 L flask of 2xYT + 0.1% glucose + Amp. Grow the culture at 180 rpm at 37℃ shaker until $OD_{600}$ of 0.6, then, switch to shaking at 19℃ for 1 hr. Next, induce with 0.4 mM IPTG. Shake at 180 rpm at 19℃ for 18–20 hr. Spin 1L cultures down at 3500 rpm in large Beckman Centrifuge at 4℃ for 20 min in (8.1 rotor). Discard media and gently resuspend cell pellet in ice-cold 20 ml in Buffer 1 (0.2 M Tris pH 8.0, 0.5 mM EDTA, 0.5 M Sucrose) on ice. Transfer the resuspended cells from step 2 into two smaller JLA 25.5 centrifuge tubes. Add 20 mL of ice cold

ddH$_2$O with 2x protease inhibitor cocktail (Roche Complete Ultra, Millipore Sigma 5056489001) from step 3 to the resuspended pellets. Incubate at on ice for 1 hr occasionally swirling samples gently. Spin periplasmic fractions at 13,000 x g for 15 min, 4°C, rotor 25.50. Wash 500 µL Ni resin (Qiagen, Ni-NTA, 30210) per periplasmic fraction four times in Buffer 2 (50 mM Tris pH 8.0, 250 mM NaCl). Add MgCl$_2$ and imidazole to a final concentration of 10 mM to each periplasmic fraction. Add beads to periplasmic fractions and nutate at 4°C for 2 hr. Spin down beads at 2000 x g, 10 min, 4°C. Transfer beads either to a 50-mL gravity column. Wash the beads with 20 column volumes of Buffer 3 (50 mM Tris pH 8.0, 500 mM NaCl, 20 mM Imidazole). Elute protein with three column volumes of Buffer 4 (50 mM Tris pH 8.0, 500 mM NaCl, 300 mM Imidazole). Analyze eluate by SDS-PAGE 4–12% Invitrogen (Invitrogen, NP0321PK2). Fabs as two bands run around 30 kDa in reducing conditions, or 50 kDa in non-reducing conditions. Dialyze eluate O/N in Dialysis cassette 10 kD molecular weight cutoff at 4°C against 150 mM KOAc, 20 mM HEPES pH 6.8.

## Purification of overexpressed yeast EMC5-3xflag

The OE-Emc5-3xflag yeast strain were grown in YEPD media in a 40 L fermenter, harvested and flash frozen in liquid nitrogen. Cell pellets were thawed and diluted in lysis buffer (50 mM HEPES pH 6.8, 150 mM KOAc, 2 mM MgOAc, 1 mM CaCl2, 0.2M Sorbital, 2x Protease Inhibitor). Bead beating (10 times → 1 min on, 2 min off) was used to lyse cells. For 25 g of cells, 0.1 mm cold beads were added and lysis buffer up to the top of the 50 mL canister. After lysis, beads were filtered and solution centrifuged at 10,000 xg for 10 min. Supernatants were ultracentrifuged at 42,000 RPM (Ti 45 rotor) for 2 hr. Supernatant was discarded. Membrane pellet was combined with the lipid layer, and resuspended in lysis buffer and then a precooled dounce homogenizer was used to dounce 20 times. Membranes were aliquoted and flash frozen in liquid nitrogen. On ice, 150 mL of solubilization buffer (50 mM HEPES pH 6.8, 150 mM KOAc, 2 mM MgOAc, 1 mM CaCl2, 15% glycerol, 1% b-DDM, 2x Protease Inhibitor) was added incrementally to 7.5 g of thawing membranes, nutated at 4°C for 1 hr in JA 25.5 rotor tubes, and centrifuged at 20,000 rpm for 45 min. Meanwhile 2.5 mL of αFLAG agarose beads (Millipore A2220) were rinsed in 50 mL of low-salt buffer (50 mM HEPES pH 6.8, 150 mM KOAc). Supernatant was added to αFLAG beads and nutated at 4°C for 2 hr. Resulting solution was applied over a glass column. After flowing through unbound solution, αFLAG beads were washed with 100 mL low-salt buffer, 100 mL high-salt buffer (50 mM HEPES pH 6.8, 300 mM KOAc, 0.05% b-DDM), and 100 mL low-salt buffer. αFLAG beads were resuspended in 10 mL of low-salt buffer and 300 µL of TEV (1.15 mg/mL) was added and nutated overnight at 4°C. Removed supernatant from beads by low-speed spin and applied over 500 µL of NiNTA beads equilibrated with low-salt buffer to remove excess TEV. Flow through glass column and collect supernatant. Using a 100 kD concentrator (Millipore, UFC910008) solution was concentrated to 2 mg/mL. Concentrated EMC protein was applied to the Akta Explorer Superose 6 Increase column (Cytiva, 29091596) for size exclusion chromatography in the size exclusion buffer (20 mM HEPES pH 6.8, 150 mM KOAc, 0.05% b-DDM). Fractions were evaluated by SDS-PAGE Coomassie stain and negative stain electron microscopy then EMC peak fractions were pooled and incubated with 2x molar excess of Fab, either Fab DH4 or Fab DE4, for 30 min on ice. Solution was applied to Akta Explorer Superose 6 Increase for size exclusion of Fab bound EMC. Resulting EMC-Fab fractions were evaluated by SDS-PAGE Coomassie stain and EMC-Fab peak fractions were pooled.

## Purification and nanodisc reconstitution of endogenous yeast EMC5-3xflag

Yeast was grown in rich media (YPAD) in a 65L fermenter until OD 2.6. Cell pellets were harvested and flash frozen in liquid nitrogen. Pellets were ground using three cycles in a French press. As above, the resulting solution was ultracentrifuged to separate membranes, dounced to homogenize, and flash frozen in liquid nitrogen. Thawed membranes were solubilized in 1% b-DDM (Anatrace, D310) nutating at 4°C for 1 hr then centrifuged to separate solubilized membranes from the pellet. Supernatant was applied to equilibrated αFLAG beads, nutated at 4°C for 1 hr, and applied over a disposable plastic column at 4°C. αFLAG beads were washed with low-salt buffer and high salt buffer. Then washed with low-salt buffer with b-DDM+CHS (Anatrace, CH210) (10:1) in place of b-DDM. αFLAG beads were then transferred to a 15-mL Eppendorf tube for TEC cleavage and nanodisc reconstitution.

Bio-Beads SM-2 (Bio-Rad) were prepared ~400 µL biobeads, rinsing with EtOH, and then water four times. Yeast Extract Total (Avanti Polar Lipids, 190000 C-100mg) was prepared by transferring chloroform resuspended solution to a glass vial, drying the lipids into a film with nitrogen gas, drying in a vacuum desiccator overnight, and then solubilizing the lipids first in water and then in size exclusion buffer with DDM+CHS by bath sonication, aliquots stored at −20°C until use. A total of 200 µL of TEV protease (5 mg/mL) and 150 µL of 1 mg/mL Yeast Total Extract solubilized in b-DDM+CHS, at room temperature for 30 min. Then added MSP1D1, purified as described previously (*Ritchie et al., 2009*), to a ratio of 200:10:1 (Yeast total extract:MSP1D1:EMC), at 4°C for 10 min. Then activated Bio-Beads SM-2 (Bio-Rad), ~300 µL, were added and nutated overnight. On-bead reconstitution employed adapted from *Laverty et al., 2019*. In the morning, ~100 µL more Bio-Beads SM-2 (Bio-Rad) were added and 2x molar excess of FabDH4, nutated for another hour. Beads and solution applied to an EconoPac column (Bio-Rad). Flow through was collected and solution was applied to a 100 kD (Amicon) concentrator. Resulting concentrated EMC was applied to the Akta Explorer Superose 6 Increase column for size exclusion chromatography. Peak fractions were pooled for SDS-PAGE Coomassie stain, negative stain, and cryo-EM evaluation. Key reagents used are provided in the Key Resource Table.

## Cryo-EM sample preparation and data collection for yEMC
### Overexpressed EMC + Fab DE4 in b-DDM
Following size exclusion sample was prepared for cryo-electron microscopy. A total of 3 µL of sample (0.1 mg/mL EMC + Fab DE4 in 20 mM HEPES pH 6.8, 150 mM KOAc, 0.05% b-DDM) was applied to the grid, incubated for 10 s, then blotted with no offset for 6.5 s and plunge frozen in liquid ethane using a Vitrobot Mark III at 5°C, Whatman #1 filter paper, and 100% humidity. Protein was frozen on glow discharged Ultrathin Carbon Film on a Lacey Carbon Support Film (Ted Pella, 01824) and stored under liquid nitrogen until imaging. This dataset was collected on the 300 kV Technai Polara at UCSF with a 30 µm C2 aperture, 100 µm Objective aperture, and K2 Summit detector operated in super-resolution mode. 1536 micrographs were collected using SerialEM (*Mastronarde, 2005*) at a magnification of 31,000X (0.6078 Å/ super resolution pixel) as dose-fractionated stacks of 40 frames x 0.2 s exposures (1.42 e$^-$/Å$^2$) for a total dose of ~56.85 e$^-$/Å$^2$ (*Table 1*).

### Overexpressed EMC + Fab DH4 in b-DDM
Following size exclusion sample was prepared for cryo electron microscopy. A total of 3 µL of sample (0.1 mg/mL EMC + Fab DH4 in 20 mM HEPES pH 6.8, 150 mM KOAc, 0.05% b-DDM) was applied to the grid, incubated for 10 s, then blotted with no offset for 7 s and plunge frozen in liquid ethane using a Vitrobot Mark III at 4°C, Whatman #1 filter paper, and 100% humidity. Protein was frozen on glow discharged Ultrathin Carbon Film on a Lacey Carbon Support Film (Ted Pella 01824). This dataset was collected at the HHMI Janelia Research Campus on Titan Krios 2, a 300 kV microscope equipped with a 50 µm C2 aperture, 70 µm objective aperture, and K2 Summit detector operated in super-resolution mode. A total of 3357 micrographs were collected using automated SerialEM (*Mastronarde, 2005*) collection with defocus range set between −1 and −3 µm at a magnification of 22,500X (0.655 Å/ super resolution pixel) as dose-fractionated stacks of 50 frames x 0.2 s exposures (1.165 e$^-$/Å$^2$) for a total dose of ~58.3 e$^-$/Å$^2$ (see *Table 1*).

### Endogenous EMC + Fab DH4 in MSP1D1-yeast total extract nanodisc
Following size exclusion sample was prepared for cryo electron microscopy. Four µL of sample (~0.8 mg/mL EMC + Fab DH4 in nanodisc in 20 mM HEPES pH 6.8, 150 mM KOAc, 0.05% b-DDM) was applied to the grid from the left side, then blotted with no offset for 2.5 s, then another 4 µL of sample was applied to the right side of the grid (without glow discharge) and blotted for 3.5 s, and plunge frozen in liquid ethane using a Vitrobot Mark IV at 4°C, Whatman #1 filter paper, and 100% humidity. Protein was frozen on R 1.2/1.3 grids with 300 Au mesh (Quantifoil, Germany). This dataset was collected at UCSF on the Titan Krios 2, a 300 kV microscope equipped with a 70 µm C2 aperture, 100 µm objective aperture, and K3 detector operated in CDS mode. 5949 micrographs were collected using automated SerialEM (*Mastronarde, 2005*) collection with defocus range set between −0.8 and −2 µm at a magnification of 105X (0.4265 Å/ super resolution pixel) as dose-

fractionated stacks of 100 frames x 0.06 s exposures (0.67 e⁻/Å² for a total dose of ~67 e⁻/Å² (see *Table 1*).

## Image analysis and 3D reconstruction for yEMC

### Overexpressed EMC + Fab in b-DDM

Image processing schematic (*Figure 2—figure supplement 1*) and *Table 1* have additional details. All dose-fractionated image stacks were corrected for motion artefacts, 2x binned in the Fourier domain, and dose-weighted using MotionCor (*Li et al., 2013*) for the DDM datasets, resulting in one dose-weighted and one unweighted integrated image per stack with pixel sizes of 1.22 Å (DDM - Polara) or 1.31 Å (DDM – Janelia Krios). The parameters of the Contrast Transfer Function (CTF) were estimated using GCTF-v1.06 (*Zhang, 2016*) and the motion-corrected but unweighted images. For each dataset, ~1000 particles per dataset were manually selected and averaged in 2D using RELION 2.0 (*Kimanius et al., 2016*). The resulting class sums were then used as templates for auto-mated particle picking using Gautomatch-v0.55 (*Zhang, 2016*), followed by extraction in RELION 2.0. Five rounds of 2D classification were performed to eliminate ice contamination, particles near carbon edges, and 2D class without visible secondary structure features. Subsequent particles were subjected to 3D auto-refine in Relion 2.0. The Polara dataset was processed providing a reference model created in Spider (*Shaikh et al., 2008*) roughly mimicking the dimensions seen in 2D projec-tions, then a second round was run using the resulting volume before two rounds of 3D classification without alignments. The resulting subset of particles were subjected to 3D auto-refine and then 3D classification with local alignments. The best 83,599 particles were then subjected to 3D refinement resulting in a 3D volume with ~8 Å reported resolution, which was rescaled and low-pass filtered for use as the reference for the DDM Krios dataset. 3D classification without alignments, 3D refinement, 3D classification with local alignments, and 3D auto refinement were performed resulting in a ~ 7 Å structure composed of 170,186 particles. Both resulting reconstructions overlay with one another, despite having Fab DH4 in one sample and DE4 in the other. Furthermore, they both displayed a severe orientation bias, and 3D reconstructions appeared streaky.

Particles from both datasets were re-extracted and scaled to a common pixel size of 1.35 Å and box size of 266. The combined dataset was subjected to two rounds of 3D refinement to form a con-sensus structure at ~6.8 Å all conducted in Relion 2.0. These particles were then subjected to 3D refinement in THUNDER (*Hu et al., 2018*) using soft-edged mask. THUNDER produced a resulting 3D reconstruction that visually appeared less distorted along the axis of overrepresented views and resulted in a ~ 4.8 Å consensus structure. Postprocessing was done in Relion 3.0 resulting in a ~ 4.3 Å sharpened map and output was used to generate the FSC plot (*Figure 2—figure supplement 1*). Molecular graphics and analyses were performed with the UCSF Chimera package (*Pettersen et al., 2004*) and Coot 0.8.7 and Coot 0.9 (*Emsley and Cowtan, 2004*; *Emsley et al., 2010*). Local resolu-tion was computed by inputting mask and half maps into Cryosparc two local resolution (*Stagg et al., 2014*; *Punjani et al., 2017*; *Punjani et al., 2019*) and visualizing the resulting map and scaling in UCSF Chimera.

### Endogenous EMC + Fab DH4 in MSP1D1-yeast total extract nanodisc

All dose-fractionated image stacks were corrected for motion artefacts, 2x binned in the Fourier domain, and dose-weighted using MotionCor2 (*Zheng et al., 2017*) using Focus (*Biyani et al., 2017*) resulting in a 2x binned pixel size of 0.835 Å (nanodisc – UCSF Krios). The parameters of the Contrast Transfer Function (CTF) were estimated using GCTF-v1.06 (*Zhang, 2016*) and the motion-corrected but unweighted images. Data were then split into five groups of 1000 micrographs for processing until they were combined in 3D. Roughly ~1000 particles per subset were manually selected and averaged in 2D using RELION 3.0 (*Zivanov et al., 2018*) for the nanodisc dataset. The resulting class sums were then used as templates for automated particle picking using Autopick in Relion 3.0, followed by extraction and one round of 2D classification per subset to remove ice con-tamination. The resulting subsets of particles were subject to 3D refinement. Combining the RELION star files these particles were imported into Cryosparc 2.0 (*Punjani et al., 2017*; *Punjani et al., 2019*) along with a reference model. These data were subjected to non-uniform homogenous refine-ment, a round of four class 3D heterogeneous refinement, another round of non-uniform refinement for the best class (roughly 1.2 million particles), non-uniform homogeneous refinement, a round of

two class 3D heterogeneous refinement, and another non-uniform homogeneous refinement for the best class (roughly 500,000 particles). These were then exported to RELION 3.0 using PyEM (*Asarnow et al., 2019*). 3D Classification was performed with local alignments, then CTF refinement of the best class (230,528 particles) resulting in a ~ 3.2 Å final reconstruction. This was post-processed in both RELION 3.0 and using phenix.autosharpen, both resulting maps were used for model building.

## Model building and refinement of yEMC in nanodiscs

Structural biology applications used in this project were compiled and configured by SBGrid (*Morin et al., 2013*). The yeast EMC structure was built de novo using Coot (**version 0.8.7 and 0.9**) and UCSF ChimeraX (*Goddard et al., 2018*). Visible secondary structure was built by hand for the entire structure using overlays of the yEMC detergent consensus map as well as the yEMC nanodisc unsharpened and sharpened map. Starting with the best resolved transmembrane helices, sequence was placed for each of the predicted transmembrane helices, using TMHMM (*Krogh et al., 2001*), in the yEMC proteins. Visual inspection for landmark residues (tryptophan, tyrosine, leucine, and proline) in the sequences that correlated with the position of well densities as well as fit correlation in UCSF Chimera was computed to assign identities for yEMC1, yEMC3, yEMC5, and yEMC6. Connectivity between the EMC1 assigned helix to the lumenal domain was used to start assigning sequence for the lumenal portion of EMC1. Secondary structure prediction was computed for all yEMC proteins using Phyre2 (*Kelley et al., 2015*) and Quick2D, a tool within the Max-Plank Institute for Developmental Biology Bioinformatics Toolkit that visualizes several different secondary structure predictors (*Jones, 1999*; *Cuff and Barton, 2000*; *Ouali and King, 2000*; *Rost, 2001*; *Lupas et al., 1991*; *Jones et al., 1994*; *Ward et al., 2004*; *Peng et al., 2006*; *Obradovic et al., 2005*). Secondary structure prediction was used to check and guide sequence assignment of beta strands and helices. Next several homology models were computed and overlain for yEMC2, with a predicted TPR structural domain, using Robetta (*Raman et al., 2009*; *Song et al., 2013*), I-TASSER (*Zhang, 2008*; *Roy et al., 2010*; *Yang et al., 2015*), Phyre2 (*Kelley et al., 2015*), and RaptorX (*Källberg et al., 2012*). These were used in addition to secondary structure prediction to guide sequence assignment, loop building, and helical packing. Fab DH4 starting structure was computed using Phyre2 1-to-1 threading against a crystal structure of a monoclonal Fab (PDB 1M71, *Vyas et al., 2002*). EMC3, EMC5, and EMC6 were built off of the transmembrane helices using sphere refinement, real space refinement, regularization, and visual monitoring of the Ramachandran plot in Coot. EMC7 and EMC10 both form beta sandwich folds on the exterior of the EMC1 lumenal domain, beta strand sequence was placed for both in both densities, position of aromatic residues and loop length differed between the two allowing assignment of each. After building EMC1-3, EMC5-7, and EMC10, there remained several transmembrane helices and a beta strand fitted into the lumen but not connected to EMC1, EMC7, or EMC10. The resolution of the lumenal domain is better than 3 Å in most parts allowing for sequence placement of the EMC4 C-terminus and C-terminal transmembrane helix. The connectivity of the transmembrane helix to the cytoplasmic domain was not resolved. However, there was an additional poorly resolved short helix and loop density in the cytoplasmic domain which was assigned to EMC4.

Two poorly resolved transmembrane helices remained, however, due to the fact they did not have clear connectivity to any built strand, poly alanine alpha helices were built in but not assigned to a yEMC protein (*Figure 5—figure supplement 5*). EMC4 had density in the cytoplasmic domain as well as the lumenal domain, suggesting that it has either one or three transmembrane passes. EMC7 and EMC10 were predicted to have transmembrane helices, however, the connection between the lumenal densities and those predicted transmembrane helices was not clear. Additional density that was not built into was visualized in UCSF ChimeraX (*Goddard et al., 2018*) and allowed for subsequent assignment of several glycosylated residues and one POPC molecule. Each subunit was built in a separate pdb file and subjected to iterative rounds of phenix.real_space_refine (*Adams et al., 2011*; *Liebschner et al., 2019*) into segmented maps preceded and followed by adjustment in Coot. Manual assignment of secondary structure restraints was used and improved during Phenix refinement. Once all of the well-resolved secondary structure was assigned to yEMC subunits, PDBs were combined and subjected to iterative rounds of phenix.real_space_refine (*Adams et al., 2011*; *Afonine et al., 2018*; *Liebschner et al., 2019*) in the unsharpened and then sharpened maps. Loops were built back where the connectivity was clear and then refined again in

Phenix and Coot. PDBs were prepared for refinement steps using phenix.reduce to add hydrogens throughout refinement steps, ReadySet to generate cif restraints, and Phenix PDB preparation tool for creating mmCIF files for deposition. Representative regions of the model as well as the map-to-model FSC can be found in *Figure 2—figure supplement 5*. Model for the yEMC nanodisc sample was used to generate reference model restraints for phenix real space refinement of yEMC DDM model.

## Cloning and expression constructs for hEMC

A modified version of the biGBac (*Weissmann et al., 2016*) multi-gene cloning method was combined with the BacMam (*Goehring et al., 2014*) mammalian expression system to allow for recombinant production of human EMC (hEMC). hEMC subunits were individually inserted into pEG, with EMC5 bearing a C-terminal Flag-tag. To amplify gene-expression cassettes (GEC) from pEG, original forward primers from biGBac were used in combination with modified reverse primers bearing complementarity downstream of the SV40 terminator sequence. GECs were inserted into pBIG1a-e vectors as follows: pBIG1a (EMC1 - Uniprot code Q8N766-1), pBIG1b (EMC4 – Q5J8M3-1; EMC5-Flag – Q8N4V1-1, which encodes DYKDDDDK immediately after R131; EMC6 – Q9BV81), pBIG1c (EMC2 – Q15006; EMC3 – Q9P0I2-1; EMC7 – Q9NPA0), pBIG1d (EMC8 – O43402-1; EMC9 – Q9Y3B6), pBIG1e (EMC10 – Q5UCC4-1). These were subsequently combined into pBIG2abcde to yield a single expression vector containing all 10 hEMC subunits. Bacmid was generated in DH10 EMBacY *E. coli* and subsequently transfected into Sf9 insect cells using FuGENE (Promega) reagent. Virus was amplified in Sf9 cells up to P3 and virus supernatant sterilized by filtration.

## hEMC expression, purification, and nanodisc reconstitution

Recombinant hEMC was expressed by baculovirus transduction of human embryonic kidney (HEK) 293S GnTI- cells grown in suspension. Cells were maintained at 37°C in Freestyle 293 Expression Medium (Thermo) and expanded with home-made suspension medium (*Chaudhary et al., 2012*) in 2 L shaker flasks. For expression of hEMC, 10% (v/v) P3 virus was added to 800 mL of HEK culture at a cell density $>3\times10^6$. 16 hr post-transduction, 10 mM butyrate was added and the temperature reduced to 30°C. Cells were harvested 48 hr later and stored frozen at −80°C.

For purification, 15–20 g of cell pellet was thawed and resuspended in 60–80 mL Lysis Buffer containing 50 mM ammonium citrate pH 6.0, 150 mM sodium chloride, 0.001 mg/mL Benzonase, EDTA-free protease inhibitor cocktail (1 tablet per 50 mL of buffer), and lysed by Dounce homogenization on ice (50 strokes). Glyco-diosgenin (GDN, Anatrace) was added to the lysate at 2% (w/v) and cellular membranes solubilized for 3 hr at 4°C under constant stirring. Insolubilized material was removed by centrifugation at 100,000 x g, supernatant incubated with 2 mL M2 Flag-affinity resin in-batch for 2 hr at 4°C. The resin was poured into a column and unbound proteins washed away with 25 column volumes (CV) of Wash Buffer containing 20 mM ammonium citrate pH 6.0, 150 mM sodium chloride, 0.01% (w/v) GDN. Bound hEMC was eluted in 10 CV Wash Buffer containing 0.3 mg/mL Flag peptide and concentrated to <500 μL using centrifugal concentration filters with 100 kDa cut—off (Amicon). Sample was polished using size-exclusion chromatography (SEC) on a Superose 6 Increase 10/300 GL column (GE Healthcare) with Running Buffer containing 10 mM ammonium citrate pH 6.0, 100 mM sodium chloride, 0.25 mM TCEP, 0.01% (w/v) GDN. Peak fractions containing hEMC were pooled, concentrated to ~3 mg/mL and used immediately for cryo-EM grid preparation. hEMC in nanodiscs composed of MSP1D1 scaffold protein and 1-palmitoyl-2-oleoyl-sn-glycero-3-phosphatidylcholine (POPC) was reconstituted following Flag-affinity chromatography. The MSP1D1 expression vector was a gift from Franz Hagn (TUM, Germany) and the scaffold protein purified from *E. coli* following a published protocol (*Hagn et al., 2018*). Prior to reconstitution, hEMC purified by Flag-affinity chromatography was mixed with MSP1D1 and POPC (solubilized as 25 mM stock in 5% n-dodecyl β-D-maltoside) in a 1:4:50 ratio and this mixture incubated on ice for 2 hr. Nanodisc reconstitution was achieved by incubation with 0.5–1 mL Bio-Beads SM-2 (Bio-Rad) for 16 hr at 4°C under constant rotation. The liquid phase was aspirated, concentrated to <500 μL and injected onto a Superose 6 s column with buffer containing 10 mM ammonium citrate pH 6.0, 100 mM sodium chloride, 0.25 mM TCEP, to separate nanodisc-embedded hEMC from empty nanodiscs. Peak fractions were pooled and concentrated to ~2 mg/mL for immediate cryo-EM grid preparation.

## Cryo-EM sample preparation and imaging for hEMC

Four µL of freshly purified hEMC (in detergent or nanodisc) was applied to glow discharged copper Quantifoil holey carbon grids (R1.2/1.3 300 mesh) at 100% humidity and 4°C in a Vitrobot Mark IV (Thermo) and incubated for 30 s. Excess liquid was blotted away with filter paper (blot force 4–6, blot time 4 s) and the grid plunge-frozen into liquid ethane. Samples were imaged on a FEI Titan Krios microscope operating at 300 kV, equipped with a post-column GIF and a K3 direct detector operating in counting mode. Images were recorded at a nominal magnification of 105,000x (0.8512 Å/pixel at the specimen level) for hEMC in nanodiscs or 81,000x (1.094 Å/pixel at the specimen level) for hEMC in detergent, with target defocus ranging between 0.7 and 2.8 µm and total exposure of ~70 e/Å$^2$ using SerialEM (*Mastronarde, 2005*). On-the-fly motion correction, CTF estimation and templated particle auto-picking were performed using a pipeline implemented in Focus (*Biyani et al., 2017*).

## Cryo-EM data processing for hEMC in detergent

Preprocessing in Focus included dose-weighted motion correction using Motioncor2 (*Zheng et al., 2017*), CTF estimation using Gctf (*Zhang, 2016*) and templated autopicking using Gautomatch (*Zhang, 2016*). The autopicking template originated from a reconstruction of hEMC in GDN micelles, with data acquired on a K2 (Gatan) direct electron detector (operated in counting mode) under liquid nitrogen conditions using a Glacios microscope (Thermo) operated at 200 kV. 3713 micrographs with a maximal resolution estimate better than 5 Å were imported into Relion 3.0 (*Zivanov et al., 2018*), from which ~ 3.35 million particles were extracted applying fourfold binning. These were subjected to three rounds of 2D classification and two rounds of 3D classification (using the reconstruction obtained from the 200kV dataset as reference), followed by 3D autorefinement. This reconstruction was used as initial model for three rounds of 3D classification of the original ~3.35 million particles (first round: K = 10, T = 10; second round: K = 10, T = 10; third round: K = 3, T = 16), yielding a set of 144,222 particles. This set was re-extracted at full pixel size, followed by masked 3D autorefinement, producing a reconstruction at 3.77 Å overall resolution. Application of non-uniform refinement in cryoSPARC (*Punjani et al., 2017*; *Punjani et al., 2019*) further improved the map quality and overall resolution to 3.60 Å.

## Cryo-EM data processing for hEMC in nanodiscs

Micrographs were preprocessed using Focus in a similar manner as for hEMC in detergent. 9164 micrographs with a maximal resolution estimate better than 5 Å were imported into Relion 3.0, from which ~ 5.9 million particles were extracted applying fourfold binning. These were subjected to three rounds of 3D classification (using hEMC in GDN as reference for the first round), after which 386739 particles were kept and re-extracted to full pixel size. Particles were aligned using global angular search 3D classification (K = 1, T = 4) before one further round of 3D classification with a soft mask and skipping alignment (K = 6, T = 8), to isolate a set of 177560 homogeneous hEMC particles. Masked 3D autorefinement of this particle set yielded a map at 3.6 Å overall resolution. Implementation of cryoSPARC non-uniform refinement led to a consensus map at 3.4 Å global resolution. To aid de novo model building of cytoplasmic and luminal domains, these parts were subjected to masked focused classification (K = 5, T = 8), 3D autorefinement and post-processing in Relion, yielding improved maps at 3.4 Å and 3.2 Å, respectively. To obtain highest quality maps of the transmembrane domains, the 177560 particles from consensus refinement were processed using Sidesplitter (*Ramlaul et al., 2020*), producing a 3.3 Å global map after Relion post-processing, where transmembrane helix pitch and side chains were well resolved and allowed for unambiguous sequence assignment. The final particle set was further subjected to 3D variability analysis (*Punjani and Fleet, 2020*) in cryoSPARC, revealing the presence or absence of the EMC7 lumenal domain between the EMC1 beta-propellers. Heterogeneous refinement, using a map from 3D variability analysis containing stronger EMC7 density as reference, allowed for further sub-classification of the consensus particle set. Non-uniform refinement of the class containing stronger EMC7 density produced a map at 3.5 Å global resolution, which was subsequently used to build an EMC7 model.

## Model building and refinement of hEMC in nanodiscs and detergent

Given the higher quality hEMC nanodisc map compared to the detergent map, the former was used for de novo model building in Coot (*Emsley and Cowtan, 2004*; *Emsley et al., 2010*). Focused luminal and cytoplasmic, as well as Sideplitter maps, permitted assignment of amino acid sequence throughout all parts of hEMC. Inspection of structural homology and secondary structure predictions for the hEMC subunits produced via HHpred and Quick2D servers (*Zimmermann et al., 2018*) predicted the luminal domain of EMC1, the largest hEMC subunit, to consist of two beta-propellers. EMC7 and EMC10 are predicted to feature beta-sandwich structures in the lumen. A final missing beta-strand of the EMC1 membrane proximal propeller could be assigned to the luminal C-terminus of EMC4, which forms a parallel sheet with EMC1 residues 668–674. Almost all EMC2 is predicted to form an alpha-solenoid structure harboring several TPR motifs. Analysis of EMC8 and EMC9 amino acid sequences revealed structural homology to CSN5 (deneddylase subunit of the CSN complex) and Rpn11 (deubiquitinase subunit of the 19S proteasomal regulatory particle) peptide hydrolase folds. The globular density sitting on the distal face of the EMC2 solenoid, facing away from the rest of the complex, was modeled with the EMC8 sequence, which shares ~45% amino acid sequence identity with EMC9. Additional helical density sitting sideways on top of the EMC2 solenoid could be modeled as two cytoplasmic helices of EMC3 as well as the extended, partially helical meander of the EMC3 C-terminus. Beta-strand-like density on the EMC8 surface, commonly occupied by deubiquitinase substrate peptides, was assigned to the extreme N-terminus of EMC4, with a further downstream part of this cytoplasmic domain snaking along EMC2 and EMC3 toward the transmembrane part of hEMC.

Clear side-chain resolution and excellent connectivity of the Sidesplitter map, within the nanodisc encircled membrane domain, allowed us to model all predicted transmembrane helices of EMC1, EMC3, EMC5, and EMC6. EMC5 extends its C-terminus outside the membrane, which snakes through the central cavity of the EMC2 solenoid on the cytoplasmic side. Inspection of the map at lower thresholds revealed density for at least two additional transmembrane helices facing EMC3 and EMC6 on one side of the complex: continuous density from one of these helices toward the luminal EMC4 C-terminus indicates that at least one of these gate helices represent EMC4's C-terminal transmembrane helix. However, given poor map resolution and connectivity in this region, we left the other gate helices unassigned.

Model refinement was performed using real-space refinement in Phenix (*Adams et al., 2011*), applying secondary structure and Ramachandran restraints. Initially, luminal and cytoplasmic domains were refined individually against their focused maps, after which the improved models were rigid-body placed and refined against the non-uniform refined consensus map. The transmembrane domain was likewise first refined against the Sidesplitter map, after which all parts of hEMC were combined into a consensus model and refined against the consensus map.

The refined hEMC nanodisc model was subsequently docked into the hEMC detergent map, revealing a relative rotation of the entire lumenal domain. The fitted model was manually adjusted in Coot (*Emsley and Cowtan, 2004*; *Emsley et al., 2010*) and refined using Phenix real-space refinement (*Adams et al., 2011*). Different masking strategies failed to produce stronger density for the EMC7 lumenal domain in the hEMC detergent maps, despite EMC7 levels being comparable to the other hEMC subunits in subsequent mass spectrometry analysis. EMC7 thus remains absent from our hEMC detergent model, perhaps due to conformational heterogeneity.

## Mass spectrometric analysis of purified hEMC samples in detergent or nanodiscs

GDN solubilized or nanodisc reconstituted hEMC purified by Flag-affinity chromatography and SEC was subjected to mass spectrometric analysis to assess hEMC subunit abundance. For reduction and alkylation of the proteins, proteins were incubated with SDC buffer (1% Sodiumdeoxycholate, 40 nmM 2-Cloroacetamide (Sigma-Aldrich), 10 mM tris(2-carboxyethyl) phosphine (TCEP; PierceTM, Thermo Fisher Scientific) in 100 mM Tris, pH 8.0) for 20 min at 37℃. Before digestion, the samples were diluted 1:2 with MS grade water (VWR). Samples were digested overnight at 37℃ with 1 µg trypsin (Promega).

The solution of peptides was then acidified with Trifluoroacetic acid (Merck) to a final concentration of 1% and a pH value of <2, followed by purification via SCX StageTips (*Rappsilber et al.,*

*2007*) washed with 1% TFA in Isopropanol, followed by a second wash with 0.2% TFA, eluted as one fraction with 80% Acetonitrile and 5% Ammonia (Merck). Samples were vacuum dried and re-suspended in 6 µl of Buffer A (0.1% Formic acid (Roth) in MS grade water (VWR)).

Purified and desalted peptides were loaded onto a 15-cm column (inner diameter: 75 µm; packed in-house with ReproSil-Pur C18-AQ 1.9-µm beads, Dr. Maisch GmbH) via the autosampler of the Thermo Easy-nLC 1000 (Thermo Fisher Scientific) at 50℃. Using the nanoelectrospray interface, eluting peptides were directly sprayed onto the benchtop Orbitrap mass spectrometer Q Exactive HF (Thermo Fisher Scientific).

Peptides were loaded in buffer A (0.1% (v/v) Formic acid) at 250 nL/min and percentage of buffer B (80% Acetonitril, 0.1% Formic acid) was ramped to 30% over 45 min followed by a ramp to 60% over 5 min then 95% over the next 5 min and maintained at 95% for another 5 min. The mass spectrometer was operated in a data-dependent mode with survey scans from 300 to 1650 m/z (resolution of 60000 at m/z = 200), and up to 10 of the top precursors were selected and fragmented using higher energy collisional dissociation (HCD with a normalized collision energy of value of 28). The MS2 spectra were recorded at a resolution of 15000 (at m/z = 200). AGC target for MS and MS2 scans were set to 3E6 and 1E5, respectively, within a maximum injection time of 100 and 60 ms for MS and MS2 scans, respectively. Dynamic exclusion was set to 30 ms.

Raw data were processed using the MaxQuant computational platform (*Cox and Mann, 2008*) with standard settings applied. Shortly, the peak list was searched against the reviewed human Uniprot database with an allowed precursor mass deviation of 4.5 ppm and an allowed fragment mass deviation of 20 ppm. MaxQuant by default enables individual peptide mass tolerances, which was used in the search. Cysteine carbamidomethylation was set as static modification, and methionine oxidation and N-terminal acetylation as variable modifications. The iBAQ algorithm was used for calculation of approximate abundances for the identified proteins (*Schwanhäusser et al., 2011*) which normalizes the summed peptide intensities by the number of theoretically observable peptides of the protein.

## Sequence alignments

T-coffee PSI-Coffee extension (*Notredame et al., 2000*) was used to compute sequence alignments between yEMC, hEMC, and homologous proteins (*Figure 1—figure supplements 6–7*, *Figure 3— figure supplement 3*, *Figure 5—figure supplement 3*). Outputs of these alignments were visualized in Jalview (*Waterhouse et al., 2009*) for figure creation and colored by ClustalX convention.

## Figure and video creation

All figures were assembled and edited in Adobe Illustrator. *Figure 1* and *Figure 1—figure supplement 4* were created using BioRender. All the visualization, structure figures, and structure videos were made using UCSF ChimeraX 1.0 (*Goddard et al., 2018*) and UCSF Chimera 1.14 (*Pettersen et al., 2004*). Flow cytometry plots were generated in Python and labeled in Adobe Illustrator.

## Acknowledgements

We thank J Weibazahn, P Walter, R Irannejad, J Gestwicki, R Scheltema, Ö Karayel, H Nguyen, I Johnson, N Talledge, L Kenner, E Thompson, K Hickey, J Kellermann, S von Gronau, M Feige, K Swain, M Liao, C-W Lee, F Wilfling, and members of the Weissman, Frost, and Schulman laboratories for assistance and helpful discussion; L Metzger, Z Roe-Zurz, M Tessema, DW Chester, and S Aller for assisting with fermentation; M Sun, H Autzen, and E Green for advice on nanodisc reconstitution; F Hagn and I Goba for the gift of the MSP1D1 vector and advice on nanodisc reconstitution; P Thomas and D Asarnow for computational support; M Braunfeld, G Gilbert, E Tse, D Bulkley, M Harrington, A Myasnikov and Z Yu of the UCSF Center for Advanced CryoEM for microscopy support and funded by NIH grants S10OD020054 and 1S10OD021741; J Baker-LePain and the QB3 shared cluster (NIH grant 1S10OD021596-01) for computational support; and the Howard Hughes Medical Institute (HHMI); Z Yu and H Chou of the CryoEM Facility at the HHMI Janelia Research Campus (NIH grant 1S10OD021596-01). D Bollschweiler, T Schäfer and the cryo-EM facility at the Max Planck Institute of Biochemistry; B Steigenberger, the mass spectrometry core facility at the Max Planck Institute of Biochemistry; E Gouaux for the gift of the pEG vector; A Titan X Pascal used for this

research was donated by the NVIDIA Corporation. This study was supported in part by the HDFCCC Laboratory for Cell Analysis Shared Resource Facility through a grant from NIH (P30CA082103). Molecular graphics and analyses performed with UCSF ChimeraX, developed by the Resource for Biocomputing, Visualization, and Informatics at the University of California, San Francisco, with support from National Institutes of Health R01-GM129325 and the Office of Cyber Infrastructure and Computational Biology, National Institute of Allergy and Infectious Diseases. UCSF Chimera is developed by the Resource for Biocomputing, Visualization, and Informatics at the University of California, San Francisco (supported by NIGMS P41-GM103311).

## Additional information

### Funding

| Funder | Grant reference number | Author |
|---|---|---|
| Deutsche Forschungsgemeinschaft | Leibniz Prize SCHU 3196/1-1 | Brenda A Schulman |
| Max Planck Society | | Brenda A Schulman |
| National Institutes of Health | P50AI150476 | Charles S Craik<br>Natalia Sevillano |
| National Institutes of Health | 1P41CA196276-01 | Charles S Craik |
| Helen Hay Whitney Foundation | | Matthew J Shurtleff |
| Peter und Traudl Engelhorn Stiftung | | Bastian Bräuning |
| Jane Coffin Childs Memorial Fund for Medical Research | | Nicole T Schirle Oakdale |
| National Institutes of Health | 1DP2OD017690-01 | Adam Frost |
| National Institutes of Health | GM24485 | Robert M Stroud |
| Howard Hughes Medical Institute | | Jonathan S Weissman |
| Chan Zuckerberg Initiative | | Adam Frost |
| Howard Hughes Medical Institute | 55108523 | Adam Frost |

The funders had no role in study design, data collection and interpretation, or the decision to submit the work for publication.

### Author contributions

Lakshmi E Miller-Vedam, Katerina D Popova, Conceptualization, Data curation, Formal analysis, Validation, Investigation, Visualization, Methodology, Writing - original draft, Writing - review and editing; Bastian Bräuning, Conceptualization, Data curation, Formal analysis, Funding acquisition, Validation, Investigation, Visualization, Methodology, Writing - original draft, Writing - review and editing; Nicole T Schirle Oakdale, Conceptualization, Data curation, Formal analysis, Funding acquisition, Validation, Methodology, Writing - review and editing; Jessica L Bonnar, Data curation, Formal analysis, Validation, Visualization, Methodology, Writing - review and editing; Jesuraj R Prabu, Data curation, Validation, Methodology, Writing - review and editing; Elizabeth A Boydston, Natalia Sevillano, Data curation, Formal analysis, Methodology, Writing - review and editing; Matthew J Shurtleff, Funding acquisition, Validation, Writing - review and editing; Robert M Stroud, Conceptualization, Resources, Supervision, Funding acquisition, Methodology, Writing - review and editing; Charles S Craik, Conceptualization, Supervision, Funding acquisition, Methodology, Writing - review and editing; Brenda A Schulman, Adam Frost, Jonathan S Weissman, Conceptualization, Supervision, Funding acquisition, Validation, Writing - review and editing

## Author ORCIDs

Lakshmi E Miller-Vedam (iD) https://orcid.org/0000-0002-2980-7479
Bastian Bräuning (iD) https://orcid.org/0000-0002-7194-2500
Katerina D Popova (iD) https://orcid.org/0000-0002-3927-1284
Jessica L Bonnar (iD) https://orcid.org/0000-0001-5531-4849
Elizabeth A Boydston (iD) http://orcid.org/0000-0001-8365-0436
Matthew J Shurtleff (iD) http://orcid.org/0000-0001-9846-3051
Charles S Craik (iD) https://orcid.org/0000-0001-7704-9185
Brenda A Schulman (iD) https://orcid.org/0000-0002-3083-1126
Adam Frost (iD) https://orcid.org/0000-0003-2231-2577
Jonathan S Weissman (iD) https://orcid.org/0000-0003-2445-670X

## Decision letter and Author response

Decision letter https://doi.org/10.7554/eLife.62611.sa1
Author response https://doi.org/10.7554/eLife.62611.sa2

---

# Additional files

## Supplementary files

• Supplementary file 1. Mass spectrometry analysis on purified hEMC. SEC purified hEMC in detergent (sheet 1) or nanodiscs (sheet 2) were subjected to tryptic digestion and mass spectrometry. The tables list identified proteins sorted by iBAQ score (descending order). EMC subunits are highlighted in yellow.

• Supplementary file 2. Statistical significance values for flow cytometry data. Table listing p-values for membrane controls (Sheet 1; relates to *Figure 1—figure supplement 3*) and flow cytometry for each of the three client reporters (Sheets 2, 3 and 4; relates to Main *Figures 3* and *5–7* and figures supplements to those figures).

• Supplementary file 3. Comparison of EMC point mutant effects on client proteins. Table listing point mutagenesis performed on hEMC and yEMC and assayed against different client types.

• Supplementary file 4. Uncropped western blots. Blots provided here without cropping, related to *Figure 1—figure supplements 5–6*.

• Supplementary file 5. Plasmid sequences for hEMC mutants and reporters. Table listing sequences of point mutagenesis plasmids used in the hEMC functional assay in this study.

• Transparent reporting form

## Data availability

All data generated or analyzed during this study are included in the manuscript or available at an appropriate public data repository. Flow cytometry data and analysis code is available at Github (https://github.com/katerinadpopova/emcstructurefunction) (copy archived at https://archive.software-heritage.org/swh:1:rev:7ef1dee8de00b98b2cbda4321dc1989435c89eb4/). Electron microscopy maps are available at the EMDB (unsharpened, sharpened, half maps, FSC file) (accession codes EMDB - 11732, 11733, 23003, 23033), models at the PDB (accession codes PDB - 7ADO, 7ADP, 7KRA, 7KTX), and additional cryo-EM data at EMPIAR. Key Resource Table is included as an appendix to the main article and is referenced throughout the Methods section with relevant reagents used or generated during the course of the study allowing for replication of these or request of specific cell lines and reagents. Supplementary file 1 contains raw mass spectrometry data. Supplementary file 4 contains un-cropped western blots. Supplementary file 5 contains plasmid sequences for mutant constructs generated for this study.

The following datasets were generated:

| Author(s) | Year | Dataset title | Dataset URL | Database and Identifier |
|---|---|---|---|---|
| Miller-Vedam LE, | 2020 | Cryo-EM structure of | https://www.ebi.ac.uk/ | Electron Microscopy |

| | | | | |
|---|---|---|---|---|
| Schirle Oakdale NT, Bräuning B, Boydston EA, Sevillano N, Popova KD, Bonnar JL, Shurtleff MJ, Prabu JR, Stroud RM, Craik CS, Schulman BA, Weissman JS, Frost A | | Saccharomyces cerevisiae ER membrane protein complex bound to a Fab in DDM detergent | pdbe/entry/emdb/EMD-23033 | Data Bank, EMD-23033 |
| Miller-Vedam LE, Schirle Oakdale NT, Bräuning B, Boydston EA, Sevillano N, Popova KD, Bonnar JL, Shurtleff MJ, Prabu JR, Stroud RM, Craik CS, Schulman BA, Weissman JS, Frost A | 2020 | Cryo-EM structure of Saccharomyces cerevisiae ER membrane protein complex bound to Fab-DH4 in lipid nanodiscs | https://www.ebi.ac.uk/pdbe/entry/emdb/EMD-23003 | Electron Microscopy Data Bank, EMD-23003 |
| Bräuning B, Prabu RS, Miller-Vedam LE, Weissman JS, Frost A, Schulman BA | 2020 | Cryo-EM structure of human ER membrane protein complex in GDN detergent | https://www.ebi.ac.uk/pdbe/entry/emdb/EMD-11733 | Electron Microscopy Data Bank, EMD-11733 |
| Bräuning B, Prabu RS, Miller-Vedam LE, Weissman JS, Frost A, Schulman BA | 2020 | Cryo-EM structure of human ER membrane protein complex in lipid nanodiscs | https://www.ebi.ac.uk/pdbe/entry/emdb/EMD-11732 | Electron Microscopy Data Bank, EMD-11732 |
| Miller-Vedam LE, Schirle Oakdale NT, Bräuning B, Boydston EA, Sevillano N, Popova KD, Bonnar JL, Shurtleff MJ, Prabu RS, Stroud RM, Craik CS, Schulman BA, Weissman JS, Frost A | 2020 | Cryo-EM structure of Saccharomyces cerevisiae ER membrane protein complex bound to a Fab in DDM detergent | https://www.rcsb.org/structure/7KTX | RCSB Protein Data Bank, 7KTX |
| Miller-Vedam LE, Schirle Oakdale NT, Bräuning B, Boydston EA, Sevillano N, Popova KD, Bonnar JL, Shurtleff MJ, Prabu RS, Stroud RM, Craik CS, Schulman BA, Weissman JS, Frost A | 2020 | Cryo-EM structure of Saccharomyces cerevisiae ER membrane protein complex bound to Fab-DH4 in lipid nanodiscs | https://www.rcsb.org/structure/7KRA | RCSB Protein Data Bank, 7KRA |
| Bräuning B, Prabu RS, Miller-Vedam LE, Weissman JS, Frost A, Schulman BA | 2020 | Cryo-EM structure of human ER membrane protein complex in lipid nanodiscs | https://www.rcsb.org/structure/7ADO | RCSB Protein Data Bank, 7ADO |
| Bräuning B, Prabu RS, Miller-Vedam LE, Weissman JS, Frost A, Schulman BA | 2020 | Cryo-EM structure of human ER membrane protein complex in GDN detergent | https://www.rcsb.org/structure/7ADP | RCSB Protein Data Bank, 7ADP |

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
