## [Decision Letter]

**Acceptance summary:**

This manuscript describes the cryo-EM structures of hEMC and yEMC complexes in DDM as well as in nanodiscs. These complexes are important for the integration of membrane proteins in the membrane. By combining structure determination with functional assays using three different fluorescent client reporters the authors dissect the role of EMC in the insertion of these different substrate types and significantly enhance our knowledge about the essential integration process.

**Decision letter after peer review:**

Thank you for submitting your article "Structural and mechanistic basis of the EMC-dependent biogenesis of distinct transmembrane clients" for consideration by *eLife*. Your article has been reviewed by three peer reviewers, including Volker Dötsch as the Reviewing Editor and Reviewer #1, and the evaluation has been overseen by John Kuriyan as the Senior Editor. The following individual involved in review of your submission has agreed to reveal their identity: Matthew E Call (Reviewer #2).

The reviewers have discussed the reviews with one another and the Reviewing Editor has drafted this decision to help you prepare a revised submission.

Summary:

The manuscript by Miller-Vedam et al. describes the cryo-EM structures of hEMC and yEMC complexes in DDM as well as in nanodiscs. Structure determination is state-of-the-art and the structures are of good quality. The structures are analysed and functionally interpreted using three different fluorescent client reporters to dissect the role of EMC in the insertion of these substrate types. Compared to the three previous manuscripts describing EMC structures, most aspects of the structures are analysed and the structure-based mutagenesis studies are very thorough. The depth of these analyses stands out compared to the previous publications. Therefore, the manuscript should definitely be published.

Essential revisions:

1) What is missing is a detailed discussion of how their structures and functional observations differ from the Bai (Nature), Pleiner (Science) and O'Donnell (*eLife*) studies. From a structural viewpoint, this is particularly important for the differing interpretations of the densities for the 'gate' helices – likely to be a crucial regulator of client TM access and yet the least defined region. The authors briefly mention that these interpretations are different (subsection “Structural heterogeneity suggests a role for the gate in regulating access to the insertase transmembrane cavity”) but do not discuss whether any of them are particularly consistent (or not) with their observations. Functionally, it would be useful to map the mutations made in the other studies next to their own to begin to provide a consensus view of key interactions.

2) As the other EMC manuscripts have been out now for several months before this submission, it is no longer acceptable to deal with the other EMC structures by just saying "recent cryo-EM data….published while we were in the final stages of manuscript preparation. The authors should refer to these structures in the Introduction and Discussion, and use them to support their arguments and present structural overlays with the high-resolution human and yeast structures to provide more evidence that their "maps and models are consistent with recent cryo-EM data". This would strengthen the manuscript rather than undermine it.

3) Similarly, a detailed structural superimposition of the determined yeast and human structures is missing. Figure 2 is not sufficient to see the main similarities and differences. More detail would be much appreciated.

4) The models shown in Figure 8 for the three types of substrates should be outlined in the Discussion, especially as they imply some hypotheses that are not directly supported by the data e.g. the biogenesis of polytopic membrane proteins requiring the lipid filled cavity.

5) The manuscript does not contain data proofing a holdase chaperone function – therefore the statement in the Abstract referring to "a novel non-insertase EMC function mediated by the luminal domain" should be modified.

6) Related: with respect to this suggested holdase/chaperone function other recent publications had noticed that steric hindrance would exclude a direct contact with the Sec61 channel when interacting with the ribosome. Could the authors speculate more about how the EMC regulates membrane protein folding under these steric restrictions?

---

## [Author Response]

Essential revisions:1) What is missing is a detailed discussion of how their structures and functional observations differ from the Bai (Nature), Pleiner (Science) and O'Donnell (eLife) studies. From a structural viewpoint, this is particularly important for the differing interpretations of the densities for the 'gate' helices – likely to be a crucial regulator of client TM access and yet the least defined region. The authors briefly mention that these interpretations are different (subsection “Structural heterogeneity suggests a role for the gate in regulating access to the insertase transmembrane cavity”) but do not discuss whether any of them are particularly consistent (or not) with their observations. Functionally, it would be useful to map the mutations made in the other studies next to their own to begin to provide a consensus view of key interactions.

We have now included an additional figure (Figure 2—figure supplement 9) which offers pairwise structural alignments of our hEMC and yEMC structures with the human and yeast EMC structures published by the three other labs in 2020. Structures were aligned on the core subunits EMC3 and EMC5, revealing largely superimposable cytoplasmic and transmembrane domains, while the orientation of the lumenal domains differed across the structures. Alignment RMSDs ranged from 0.85Å (between hEMC nanodiscs, this work, and hEMC nanodiscs, Pleiner et al.) to 1.19Å (between hEMC nanodiscs, this work, and yEMC detergent, Bai et al.). Comparison of functional mutant data between ours and others’ published work is addressed below.

Both our own yEMC nanodisc cryo-EM map and the yEMC map deposited by Bai et al., 2020, feature comparably more defined density for the gate helices than any hEMC map currently available (including our own human EMC maps). After close inspection of our own yEMC nanodisc map and that from Bai et al., we still firmly believe that unambiguous, map-based assignment of all three gate helices is not possible at the current resolutions (the last TM helix of EMC4 can be confidently assigned both in yEMC and hEMC models based on map continuity with the EMC4 lumenal domain). This is made more difficult since both the observed and predicted topologies of EMC4, EMC7 and EMC10 all would theoretically be compatible with contributing TM helices to the gate. This last point we have addressed in a new cartoon in Figure 5—figure supplement 5C. In the revised manuscript we have further emphasized the difficulty of unambiguous gate helix assignment using our updated set of figures in Figure 5—figure supplement 5C-E, which includes the docking of predicted EMC4, EMC7 and EMC10 TM helices into the gate helix densities.

To allow for an overview of structure-guided mutagenesis performed by our labs and others, we have compiled Supplementary file 3. This table lists point mutants and their corresponding effect on topologically distinct clients, listing all structure-guided mutants performed by others in 2020 and our own EMC structure-function studies, italisized or bolded by accumulation or depletion client reporter phenotypes.

2) As the other EMC manuscripts have been out now for several months before this submission, it is no longer acceptable to deal with the other EMC structures by just saying "recent cryo-EM data….published while we were in the final stages of manuscript preparation. The authors should refer to these structures in the Introduction and Discussion, and use them to support their arguments and present structural overlays with the high-resolution human and yeast structures to provide more evidence that their "maps and models are consistent with recent cryo-EM data". This would strengthen the manuscript rather than undermine it.

As *eLife* is aware from our communication in May (prior to the publication of the other studies), we had largely completed the structural studies but had wanted to publish a completed study with both structural analysis and the functional interrogation. We had already generated the mutants when the functional studies were interrupted due to the pandemic, and hence hoped our work would be considered for scoop protection from *eLife*. Thus, the other papers were published during the final stages of manuscript preparation.

Nonetheless, to address this, we have now expanded our comparison to the other structures and we also add references to the other recent studies in the Introduction. In the Results section we added reference to a new table (Supplementary file 3), with comparisons between the mutants in our study as well as those in the other studies , a new supplementary figure with pairwise overlays between each of the structures (Figure 2—figure supplement 9), and additional panels in the supplementary figure showing different gate helix conformations (Figure 5—figure supplement 5). While we are unable to conclude as to the identity of the transmembrane gate helices we hope the additional visual representations and discussion will make the possible models clearer.

3) Similarly, a detailed structural superimposition of the determined yeast and human structures is missing. Figure 2 is not sufficient to see the main similarities and differences. More detail would be much appreciated.

We have included an additional figure supplement (Figure 2—figure supplement 8) to show the similarities and differences between the yeast and human structures we resolved in this study using side by side surface and ribbon representations. Yeast EMC was aligned to human EMC on the core transmembrane helices (EMC3, EMC5) and tiled for comparison. Superposition of these structures as they relate to the other recent structures is also provided in Figure 2—figure supplement 9.

4) The models shown in Figure 8 for the three types of substrates should be outlined in the Discussion, especially as they imply some hypotheses that are not directly supported by the data e.g. the biogenesis of polytopic membrane proteins requiring the lipid filled cavity.

To address this comment, we have more directly discussed the model figure within the Discussion in particular as it relates to what we observe in this study and what has yet to be determined. This model figure depicts the functional assay mapped onto the structure to outline the regions that appear to be important for each of the three clients assayed. Certainly, the insertase function of the EMC has been most studied. While the largest range of effects was observed for the C-terminally inserted client reporter, SQS, we did observe several mutations that also affected the N-terminally inserted client, B1AR. Furthermore, there were a number of lumenal mutations that had no effect on B1AR but significant effects on TMEM97 and/or SQS, including several that sit at the base of the lipid-filled cavity. By contrast, apart from one at the lumenal interface of the cavity, the mutants that depleted both insertase clients in the gated cavity showed no statistically significant alteration of TMEM97 levels. Further structural and functional studies would be needed to determine where these clients directly interact with the EMC.

5) The manuscript does not contain data proofing a holdase chaperone function – therefore the statement in the Abstract referring to "a novel non-insertase EMC function mediated by the luminal domain" should be modified.

We have amended the Abstract text to better reflect the main focus of this study, which focuses on three clients as readouts of different EMC activities. Relying upon results from prior studies, we removed the word novel for the lumenal function and have added additional text in the Discussion section to better encompass prior work, current work, and knowledge gap.

6) Related: with respect to this suggested holdase/chaperone function other recent publications had noticed that steric hindrance would exclude a direct contact with the Sec61 channel when interacting with the ribosome. Could the authors speculate more about how the EMC regulates membrane protein folding under these steric restrictions?

We agree that the relationship between the translocon, the translocon-associated proteins, and the EMC is an interesting one. Our preference is to address these with experiments in the future rather than speculate. However, to address this revision we have added additional text in the discussion to speculate on possible models.